# On the Loss Landscape of Adversarial Training: Identifying Challenges and How to Overcome Them

**Chen Liu**[1]  **Mathieu Salzmann**[1]  **Tao Lin**[1]  **Ryota Tomioka**[2]  **Sabine Süsstrunk**[1]

[1] EPFL, Lausanne, Switzerland, {chen.liu, mathieu.salzmann, tao.lin, sabine.susstrunk}@epfl.ch
[2] Microsoft Research, Cambridge, UK, ryoto@microsoft.com

## Abstract

We analyze the influence of adversarial training on the loss landscape of machine learning models. To this end, we first provide analytical studies of the properties of adversarial loss functions under different adversarial budgets. We then demonstrate that the adversarial loss landscape is less favorable to optimization, due to increased curvature and more scattered gradients. Our conclusions are validated by numerical analyses, which show that training under large adversarial budgets impede the escape from suboptimal random initialization, cause non-vanishing gradients and make the model find sharper minima. Based on these observations, we show that a periodic adversarial scheduling (PAS) strategy can effectively overcome these challenges, yielding better results than vanilla adversarial training while being much less sensitive to the choice of learning rate.

## 1   Introduction

State-of-the-art deep learning models have been found to be vulnerable to adversarial attacks [18, 34, 45]. Imperceptible perturbations of the input can make the model produce wrong predictions with high confidence. This raises concerns about deep learning's deployment in safety-critical applications.

Although many training algorithms have been proposed to counter such adversarial attacks, most of them were observed to fail when facing stronger attacks [4, 10]. Adversarial training [33] is one of the few exceptions, so far remaining effective and thus popular. It uses adversarial examples generated with the attacker's scheme to update the model parameters. However, adversarial training and its variants [2, 6, 24, 42, 53] have been found to have a much larger generalization gap [37] and to require larger model capacity for convergence [49]. Although recent works [6, 40] show that the adversarial training error reduces to almost $0\%$ with a large enough model and that the generalization gap can be narrowed by using more training data, convergence in adversarial training remains much slower than in vanilla training on clean data. This indicates discrepancies in the underlying optimization landscapes. While much work has studied the loss landscape of deep networks in vanilla training [12, 13, 14, 15, 31], such an analysis in adversarial training remains unaddressed.

Here we study optimization in adversarial training. Vanilla training can be considered as a special case where no perturbation is allowed, i.e., zero adversarial budget. Therefore, we focus on the impact of the adversarial budget size on the loss landscape. In this context, we investigate from a theoretical and empirical perspective how different adversarial budget sizes affect the loss landscape to make optimization more challenging. Our analyses start with linear models and then generalize to nonlinear deep learning ones. We study the whole training process and identify different behaviors in the early and final stages of training. Based on our observations, we then introduce a scheduling strategy for the adversarial budget during training. We empirically show this scheme to yield better performance and to be less sensitive to the learning rate than vanilla adversarial training.

**Contributions.** Our contributions can be summarized as follows. 1) From a theoretical perspective, we show that, for linear models, adversarial training under a large enough budget produces a constant classifier. For general nonlinear models, we identify the existence of an abrupt change in the adversarial examples, which makes the loss landscape less smooth. This causes severe *gradient scattering* and slows down the convergence of training. 2) Our numerical analysis shows that training under large adversarial budgets hinders the model to escape from suboptimal initial regions, while also causing large non-vanishing gradients in the final stage of training. Furthermore, by Hessian analysis, we evidence that the minima reached in the adversarial loss landscape are sharper when the adversarial budget is bigger. 3) We show that a periodic adversarial scheduling (PAS) strategy, corresponding to a cyclic adversarial budget scheduling scheme with warmup, addresses these challenges. Specifically, it makes training less sensitive to the choice of learning rate and yields better robust accuracy than vanilla adversarial training without any computational overhead.

**Notation and Terminology.** We use plain letters, bold lowercase letters and bold uppercase letters to represent scalars, vectors and matrices, respectively. $\|\mathbf{v}\|$ represents the Euclidean norm of vector $\mathbf{v}$ and $[K]$ is an abbreviation of the set $\{0, 1, 2, ..., K-1\}$. In a classification problem $\{(\mathbf{x}_i, y_i)\}_{i=1}^{N}$, where $(\mathbf{x}_i, y_i) \in \mathbb{R}^m \times [K]$, the classifier consists of a logit function $f : \mathbb{R}^m \to \mathbb{R}^k$, which is usually a neural network, and a risk function $\ell : \mathbb{R}^k \times [K] \to \mathbb{R}$, which is the softmax cross-entropy loss. The adversarial budget $\mathcal{S}_\epsilon^{(p)}(\mathbf{x})$ of a data point $\mathbf{x}$, whose size is $\epsilon$, is defined based on an $l_p$ norm-based constraint $\{\mathbf{x}'|\|\mathbf{x} - \mathbf{x}'\|_p \leq \epsilon\}$, and we use $\mathcal{S}_\epsilon(\mathbf{x})$ to denote the $l_\infty$ constraint for simplicity.

Given the model parameters $\theta \in \Theta$, we use $g(\mathbf{x}, \theta) : \mathbb{R}^m \times \Theta \to \mathbb{R}$ to denote the loss function for an individual data point, ignoring the label $y$ for simplicity. If we use $\mathcal{L}_\epsilon(\theta)$ to denote the adversarial loss function under adversarial budget $\mathcal{S}_\epsilon^{(p)}(\mathbf{x})$, adversarial training solves the min-max problem

$$\min_\theta \mathcal{L}_\epsilon(\theta) := \frac{1}{N} \sum_{i=1}^{N} g_\epsilon(\mathbf{x}_i, \theta) \text{ where } g_\epsilon(\mathbf{x}_i, \theta) := \max_{\mathbf{x}'_i \in \mathcal{S}_\epsilon^{(p)}(\mathbf{x}_i)} g(\mathbf{x}'_i, \theta) . \tag{1}$$

$\mathcal{L}(\theta) := \mathcal{L}_0(\theta)$ is the vanilla loss function. If $\epsilon \neq 0$, the adversarial example $\mathbf{x}'_i$, i.e., the worst-case input in $\mathcal{S}_\epsilon^{(p)}(\mathbf{x}_i)$, depends on the model parameters. We call the landscape of functions $\mathcal{L}(\theta)$ and $\mathcal{L}_\epsilon(\theta)$ the vanilla and adversarial loss landscape, respectively. Similarly, we use $\mathcal{E}(\theta)$ and $\mathcal{E}_\epsilon(\theta)$ to represent the clean error and robust error under adversarial budget $\mathcal{S}_\epsilon^{(p)}(\mathbf{x})$. In this paper, we call a function smooth if it is $C^1$-continuous. We use $\theta_0$ to denote the initial parameters. "Initial plateau" or "suboptimal region in the early stage of training" indicate the parameters that are close to the initial ones and have similar performance. "Vanilla training" means training based on clean input data, while "vanilla adversarial training" represents the popular adversarial training method in [33].

## 2 Related Work

**Adversarial Robustness.** In this work, we focus on white-box attacks, in which the attackers have access to the model parameters. Compared with black-box attacks, white box attacks better solve the inner maximization problem in (1). In this context, [18] proposes the fast gradient sign method (FGSM) to perturb the input in the direction of its gradient: $\mathbf{x}' = \mathbf{x} + \epsilon \text{ sign}(\nabla_\mathbf{x} \mathcal{L}(\theta))$. Projected gradient descent (PGD) [33] extends FGSM by iteratively running it with a smaller step size and projecting the perturbation back to the adversarial budget. Furthermore, PGD introduces randomness by starting at a random initial point inside the adversarial budget. As a result, PGD generates much stronger adversarial examples than FGSM and is believed to be the strongest attack utilizing the network's first order information [33].

When it comes to robustness against attacks, some methods have been proposed to train provably robust models by linear approximation [5, 29, 48], semi-definite programming [36], interval bound propagation [20] or randomized smoothing [8, 39]. However, these methods either only apply to a specific type of network, have a significant computational overhead, or are unstable. Furthermore, compared with adversarial training, they have been found to over-regularize the model and significantly decrease the clean accuracy [54].

As a result, we focus on PGD-based adversarial training, which first generates adversarial examples $\mathbf{x}'$ by PGD and then uses $\mathbf{x}'$ to optimize the model parameters $\theta$. In all our experiments, the adversarial loss landscape is approximated by the loss of adversarial examples found by PGD.

**Loss Landscape of Deep Neural Networks.** Many existing works focus on the vanilla loss landscape of the objective function in deep learning. It is challenging, because the objective $\mathcal{L}(\theta)$ of a deep neural network is a high-dimensional nonconvex function, of which we only know very few properties. [26] proves the nonexistence of poor local minima for general deep nonlinear networks. [30] shows that stochastic gradient descent (SGD) can almost surely escape the saddle points and converge to a local minimum. For over-parameterized ReLU networks, SGD is highly likely to find a monotonically decreasing trajectory from the initialization point to the global optimum [38].

Furthermore, some works have studied the geometric properties of local minima in the loss landscape of neural networks. In this context, [27, 35] empirically show that sharp minima usually have larger generalization gaps than flat ones. Specifically, to improve generalization, [51] uses adversarial training to avoid converging to sharp minima in large batch training. However, the correspondence between sharp minima and poor generalization is based on empirical findings and sometimes controversial. For example, [11] shows counterexamples in ReLU networks by rescaling the parameters and claims that sharp minima can generalize as well as flat ones. Moreover, different minima of the loss function have been found to be well-connected. That is, there exist hyper-curves connecting different minima that are flat in the loss landscape [12, 14]. [55] further shows that the learned path connection can help us to effectively repair models that are vulnerable to backdoor or error-injection attacks. Recently, some methods have been proposed to visualize the loss landscape [31, 44], leading to the observation that networks of different architectures have surprisingly different landscapes. Compared with chaotic landscapes, smooth and locally near-convex landscapes make gradient-based optimization much easier.

All of the above-mentioned works, however, focus on networks that have been optimized with vanilla training. Here, by contrast, we study the case of adversarial training.

## 3 Theoretical Analysis

In this section, we conduct an analytical study of the difference between $\mathcal{L}_\epsilon(\theta)$ and $\mathcal{L}(\theta)$. We start with linear classification models and then discuss general nonlinear ones.

### 3.1 Linear Classification Models

For the simple but special case of logistic regression, i.e., $K = 2$, we can write the analytical form of $\mathcal{L}_\epsilon(\theta)$. We defer the detailed discussion of this case to Appendix A.1, and here focus on linear multi-class classification, i.e., $K \geq 3$. We parameterize the model by $\mathbf{W} := \{\mathbf{w}_i\}_{i=1}^K \in \mathbb{R}^{m \times K}$ and use $f(\mathbf{W}) = [\mathbf{w}_1^T \mathbf{x}, \mathbf{w}_2^T \mathbf{x}, ..., \mathbf{w}_K^T \mathbf{x}]$ as the logit function. Therefore, the vanilla loss function is convex as $g(\mathbf{x}, \mathbf{W}) = \log\left(1 + \sum_{j \neq y} \exp^{(\mathbf{w}_j - \mathbf{w}_y)^T \mathbf{x}}\right)$. Although $g_\epsilon(\mathbf{x}, \mathbf{W})$ is also convex, it is no longer smooth everywhere. It is then difficult to write a unified expression of $g_\epsilon(\mathbf{x}, \mathbf{W})$. So we start with the *version space* $\mathcal{V}_\epsilon$ of $g_\epsilon(\mathbf{x}, \mathbf{W})$ defined as $\mathcal{V}_\epsilon = \left\{\mathbf{W} \middle| (\mathbf{w}_i - \mathbf{w}_y)\mathbf{x}' \leq 0, \forall i \in [K], \mathbf{x}' \in \mathcal{S}_\epsilon(\mathbf{x})\right\}$.

By definition, $\mathcal{V}_\epsilon$ is the smallest convex closed set containing all solutions robust under the adversarial budget $\mathcal{S}_\epsilon(\mathbf{x})$. The proposition below states that the version space $\mathcal{V}_\epsilon$ shrinks with larger values of $\epsilon$.

**Proposition 1.** *Given the definition of the version space $\mathcal{V}_\epsilon$, then $\mathcal{V}_{\epsilon_2} \subseteq \mathcal{V}_{\epsilon_1}$ when $\epsilon_1 \leq \epsilon_2$.*

The proof of Proposition 1 is very straightforward, we put it in Appendix B.1.

In addition to $\mathcal{V}_\epsilon$, we define the set $\mathcal{T}_\epsilon$ as $\mathcal{T}_\epsilon = \left\{\mathbf{W} \middle| 0 \in \arg\min_\gamma g_\epsilon(\mathbf{x}, \gamma\mathbf{W})\right\}$. $\mathcal{T}_\epsilon$ is the set of all directions in which the optimal point is the origin; that is, the corresponding models in this direction are all no better than a constant classifier. Although we cannot write the set $\mathcal{T}_\epsilon$ in roster notation, we show in the theorem below that $\mathcal{T}_\epsilon$ becomes larger as $\epsilon$ increases.

**Theorem 1.** *Given the definition of $\mathcal{T}_\epsilon$, then $\mathcal{T}_{\epsilon_2} \subseteq \mathcal{T}_{\epsilon_1}$ when $\epsilon_1 \geq \epsilon_2$. In addition, $\exists \bar{\epsilon}$ such that $\forall \epsilon \geq \bar{\epsilon}, \mathcal{T}_\epsilon = \mathbb{R}^{m \times K}$. In this case, $\mathbf{0} \in \arg\min_{\mathbf{W}} g_\epsilon(\mathbf{x}, \mathbf{W})$.*

We defer the proof of Theorem 1 to Appendix B.2, where we also provide a lower bound for $\bar{\epsilon}$. Theorem 1 indicates that when the adversarial budget is large enough, the optimal point is the origin. In this case, we will get a constant classifier, and training completely fails.

$\mathcal{L}_\epsilon(\mathbf{W})$ is the average of $g_\epsilon(\mathbf{x}, \mathbf{W})$ over the dataset, so Theorem 1 and Proposition 1 still apply if we replace $g_\epsilon$ with $\mathcal{L}_\epsilon$ in the definition of $\mathcal{V}_\epsilon$ and $\mathcal{T}_\epsilon$. For nonlinear models like deep neural networks, these conclusions will not hold because $g_\epsilon(\mathbf{x}, \theta)$ is no longer convex. Nevertheless, our experiments in Section 4.1 evidence the same phenomena as indicated by the theoretical analysis above. Larger $\epsilon$ make it harder for the optimizer to escape the initial suboptimal region. In some cases, training fails, and we obtain a constant classifier in the end.

### 3.2 General Nonlinear Classification Models

For deep nonlinear neural networks, we cannot write the analytical form of $g(\mathbf{x}, \theta)$ or $g_\epsilon(\mathbf{x}, \theta)$. To analyze such models, we follow [43] and assume the smoothness of the function $g$.

**Assumption 1.** *The function $g$ satisfies the following Lipschitzian smoothness conditions:*

$$\begin{aligned}
\|g(\mathbf{x}, \theta_1) - g(\mathbf{x}, \theta_2)\| &\leq L_\theta \|\theta_1 - \theta_2\| \,, \\
\|\nabla_\theta g(\mathbf{x}, \theta_1) - \nabla_\theta g(\mathbf{x}, \theta_2)\| &\leq L_{\theta\theta} \|\theta_1 - \theta_2\| \,, \\
\|\nabla_\theta g(\mathbf{x}_1, \theta) - \nabla_\theta g(\mathbf{x}_2, \theta)\| &\leq L_{\theta\mathbf{x}} \|\mathbf{x}_1 - \mathbf{x}_2\|_p \,.
\end{aligned} \tag{2}$$

Based on this, we study the smoothness of $\mathcal{L}_\epsilon(\theta)$.

**Proposition 2.** *If Assumption 1 holds, then we have* [1]

$$\begin{aligned}
\|\mathcal{L}_\epsilon(\theta_1) - \mathcal{L}_\epsilon(\theta_2)\| &\leq L_\theta \|\theta_1 - \theta_2\| \,, \\
\|\nabla_\theta \mathcal{L}_\epsilon(\theta_1) - \nabla_\theta \mathcal{L}_\epsilon(\theta_2)\| &\leq L_{\theta\theta} \|\theta_1 - \theta_2\| + 2\epsilon L_{\theta\mathbf{x}} \,.
\end{aligned} \tag{3}$$

The proof is provided in Appendix B.3, in which we can see the upper bound in Proposition 2 is tight and can be achieved in the worst cases. Proposition 2 shows that the first-order smoothness of the objective function is preserved under adversarial attacks, but the second-order smoothness is not. That is to say, gradients in arbitrarily small neighborhoods in the $\theta$-space can change discontinuously.

The unsatisfying second-order property arises from the maximization operator defined in the functions $g_\epsilon$ and $\mathcal{L}_\epsilon$. For function $g_\epsilon(\mathbf{x}, \theta)$, the non-smooth points are those where the optimal adversarial example $\mathbf{x}'$ changes abruptly in a sufficiently small neighborhood. Formally, we use $\theta_1$ and $\mathbf{x}_1'$ to represent the model parameters and the corresponding optimal adversarial example. We assume different gradients of the model parameters for different inputs. If there exists a positive number $a > 0$ such that, $\forall \delta > 0$, we can find $\theta_2 \in \{\theta | \|\theta - \theta_1\| \leq \delta\}$, and the corresponding optimal adversarial example $\mathbf{x}_2'$ satisfies $\|\mathbf{x}_1' - \mathbf{x}_2'\|_p > a$, then $\lim_{\theta \to \theta_1} \nabla_\theta g_\epsilon(\mathbf{x}, \theta) \neq \nabla_\theta g_\epsilon(\mathbf{x}, \theta_1)$. $\mathcal{L}_\epsilon(\theta)$ is the aggregation of $g_\epsilon(\mathbf{x}, \theta)$ over the dataset, so it also has such non-smooth points. In addition, as the $2\epsilon L_{\theta\mathbf{x}}$ term in the second inequality of (3) indicates, the adversarial examples can change more under a larger adversarial budget. As a result, the (sub)gradients $\nabla_\theta \mathcal{L}_\epsilon(\theta)$ can change more abruptly in the neighborhood of the parameter space. That is, the (sub)gradients are more *scattered* in the adversarial loss landscape.

Figure 1 provides a 2D sketch diagram showing the non-smoothness introduced by adversarial training. The red curve represents the vanilla loss function $g(\mathbf{x}, \theta)$. Under adversarial perturbation, the loss landscape fluctuates within the light blue band. Then, the blue curve represents the worst case we can encounter in the adversarial setting, i.e., $g_\epsilon(\mathbf{x}, \theta)$. We can see that the blue curve is not smooth any more at the point where $\theta = 0$. Importantly, as the light blue band becomes wider under a larger adversarial budget, the corresponding non-smooth point becomes sharper, which means that the difference between the gradients on both sides of the non-smooth point becomes larger.

Based on Proposition 2, we show in the following theorem that the non-smoothness introduced by adversarial training makes the optimization by stochastic gradient descent (SGD) more difficult.

**Theorem 2.** *Let Assumption 1 hold, the stochastic gradient $\nabla_\theta \widehat{\mathcal{L}}_\epsilon(\theta_t)$ be unbiased and have bounded variance, and the SGD update $\theta_{t+1} = \theta_t - \alpha_t \nabla_\theta \widehat{\mathcal{L}}_\epsilon(\theta_t)$ use a constant step size $\alpha_t = \alpha = \frac{1}{L_{\theta\theta}\sqrt{T}}$ for $T$ iterations. Given the trajectory of the parameters during optimization $\{\theta_t\}_{t=1}^T$, then we can bound the asymptotic probability of large gradients for a sufficient large value of $T$ as*

$$\forall \gamma \geq 2, P(\|\nabla_\theta \mathcal{L}_\epsilon(\theta_t)\| > \gamma \epsilon L_{\theta\mathbf{x}}) < \frac{4}{\gamma^2 - 2\gamma + 4} \,. \tag{4}$$

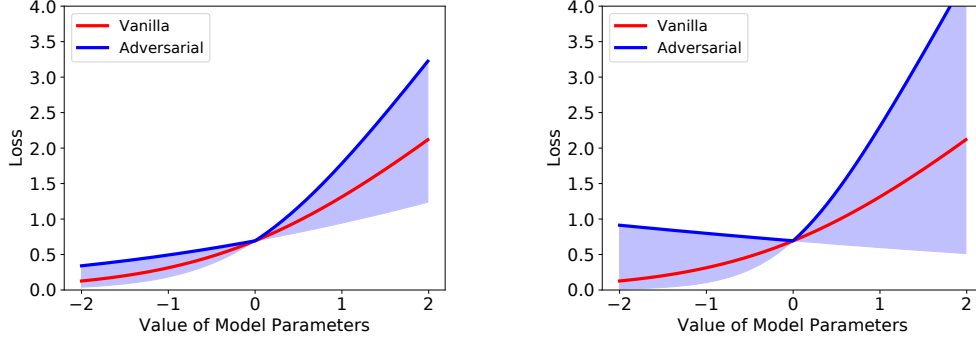

Figure 1: 2D sketch diagram showing the vanilla and adversarial loss landscapes. The clean input data $x$ is 1.0 and loss function $g(x, \theta) = \log(1 + \exp(\theta x))$. The landscape is shown in the parameter interval $\theta \in [-2, 2]$ under a small adversarial budget (left, $\epsilon = 0.6$) and a large adversarial budget (right, $\epsilon = 1.2$). The function $g_\epsilon(\mathbf{x}, \theta)$ is not smooth at $\theta = 0$.

We provide the proof in Appendix B.4. In vanilla training, $\epsilon$ is 0 and $\mathcal{L}(\theta)$ is smooth, and (4) implies that $\lim_{t \to +\infty} \|\nabla_\theta g_\epsilon(\theta_t)\| = 0$ almost surely. This is consistent with the fact that SGD converges to a critical point with non-convex smooth functions. By contrast, in adversarial training, i.e., $\epsilon > 0$, we cannot guarantee convergence to a critical point. Instead, the gradients are non-vanishing, and we can only bound the probability of obtaining gradients whose magnitude is larger than $2\epsilon L_{\theta \mathbf{x}}$. For a fixed value of $C := \gamma \epsilon L_{\theta \mathbf{x}}$ larger than $2\epsilon L_{\theta \mathbf{x}}$, the inequality (4) indicates that the probability $P(\|\nabla_\theta \mathcal{L}_\epsilon(\theta_t)\| > C)$ increases quadratically with $\epsilon$.

In deep learning practice, activation functions like sigmoid, tanh and ELU [7] satisfy the second-order smoothness in Assumption 1, but the most popular ReLU function does not. Nevertheless, adversarial training still causes *gradient scattering* and makes the optimization more difficult. That is, the bound of $\|\nabla_\theta \mathcal{L}_\epsilon(\theta_1) - \nabla_\theta \mathcal{L}_\epsilon(\theta_2)\|$ still increases with $\epsilon$, and the parameter gradients change abruptly in the adversarial loss landscape. We provide a more detailed discussion of this phenomenon in Appendix A.2, which shows that our analysis and conclusions easily extend to the ReLU case.

The second-order Lipchitz constant indicates the magnitude of the gradient change for a unit change in parameters. Therefore, it is a good quantitative metric of gradient scattering. In practice, we are more interested in the effective local Lipschitz constant, which only considers the neighborhood of the current parameters, than in the global Lipschitz constant. In this case, the effective local second-order Lipschitz constant can be estimated by the top eigenvalues of the Hessian matrix $\nabla_\theta^2 \mathcal{L}_\epsilon(\theta)$.

## 4 Numerical Analysis

In this section, we conduct experiments on MNIST and CIFAR10 to empirically validate the theorems in Section 3. Detailed experimental settings are provided in Appendix C.1. Unless specified, we use LeNet models on MNIST and ResNet18 models on CIFAR10 in this and the following sections. Our code is available on https://github.com/liuchen11/AdversaryLossLandscape.

### 4.1 Gradient Magnitude

In Section 3.1, we have shown that the training algorithm will get stuck at the origin and yield a constant classifier for linear models under large $\epsilon$. For deep nonlinear models, the initial value of the parameters is close to the origin under most popular initialization schemes [17, 22]. Although Theorem 1 is not applicable here, we are still interested in investigating how effective gradient-based optimization is at escaping from the suboptimal initial parameters. To this end, we track the norm of the stochastic gradient $\|\nabla_\theta \widehat{\mathcal{L}}_\epsilon(\theta)\|$, the robust error $\mathcal{E}_\epsilon(\theta)$ in the training set and the distance from the initial point $\|\theta - \theta_0\|$ during the first 2000 mini-batch updates for CIFAR10 models. Figure 2a, 2b, 2c evidence a clear difference between the models trained with different values of $\epsilon$. When $\epsilon$ is small, the gradient magnitude is larger, and the model parameters move faster. Correspondingly, the training error decreases faster, which means that the model quickly escapes the initial suboptimal region. By contrast, when $\epsilon$ is large, the gradients are small, and the model gets stuck in the initial region.

This implies that the loss landscape under a large adversarial budget impedes the escape from initial suboptimal plateaus in the early stage of training.

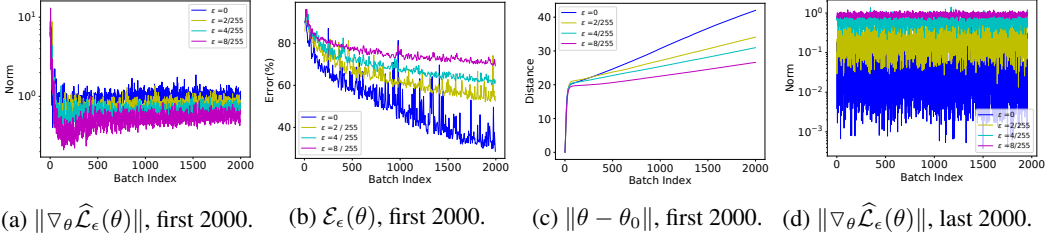

(a) $\|\nabla_\theta \widehat{\mathcal{L}}_\epsilon(\theta)\|$, first 2000.  (b) $\mathcal{E}_\epsilon(\theta)$, first 2000.  (c) $\|\theta - \theta_0\|$, first 2000.  (d) $\|\nabla_\theta \widehat{\mathcal{L}}_\epsilon(\theta)\|$, last 2000.

Figure 2: Norm of the stochastic gradient $\|\nabla_\theta \widehat{\mathcal{L}}_\epsilon(\theta)\|$, robust training error $\mathcal{E}_\epsilon(\theta)$, and distance from the initial point $\|\theta - \theta_0\|$ during the first or last 2000 mini-batch updates for CIFAR10 models.

For ReLU networks, adversarially-trained models have been found to have sparser weights and intermediate activations [9], i.e., they have more dead neurons. Dead neurons are implicitly favored by adversarial training, because the output is independent of the input perturbation. Note that training fails when all the neurons in one layer are dead for all training instances. The model is then effectively broken into two parts by this dead layer: the preceding layers will no longer be trained because the gradients are all blocked; the following layers do not depend on the input and thus give constant outputs. In essence, training is then stuck in a parameter space that only includes constant classifiers. In practice, this usually happens when the model has small width and the value of $\epsilon$ is large. This is consistent with previous findings that adversarial training needs higher model capacity [33] and too strong adversarial examples are harmful in the early stage of training [46].

Theorem 2 indicates that the gradients are non-vanishing in adversarial training and more likely to have large magnitude under large values of $\epsilon$. This is validated by Figure 2d, in which we report the norm of the stochastic gradient $\|\nabla_\theta \widehat{\mathcal{L}}_\epsilon(\theta)\|$ in the last 2000 mini-batch updates for CIFAR10 models. In vanilla training, the gradient is almost zero in the end, indicating that the optimizer finds a critical point. In this case $\|\nabla_\theta \widehat{\mathcal{L}}_\epsilon(\theta)\|$ is dominated by the variance introduced by stochasticity. However, $\|\nabla_\theta \widehat{\mathcal{L}}_\epsilon(\theta)\|$ increases with $\epsilon$. When $\epsilon$ is larger, $\|\nabla_\theta \widehat{\mathcal{L}}_\epsilon(\theta)\|$ is also larger and non-vanishing, indicating that the model is still bouncing around the parameter space at the end of training.

The decreased gradient magnitude in the initial suboptimal region and the increased gradient magnitude in the final near-minimum region indicate that the adversarial loss landscape is not favorable to optimization when we train under large adversarial budgets. Additional results on MNIST models are provided in Figure 8 of Appendix C.2.1, where the same observations can be made.

## 4.2 Hessian Analysis

To study the effective local Lipschitz constant of $\mathcal{L}_\epsilon(\theta)$, we analyze the Hessian spectrum of models trained under different values of $\epsilon$. It is known that the curvature in the neighborhood of model parameters is dominated by the top eigenvalues of the Hessian matrix $\nabla^2 \mathcal{L}_\epsilon(\theta)$. To this end, we use the power iteration method as in [51] to iteratively estimate the top 20 eigenvalues and the corresponding eigenvectors of the Hessian matrix. Furthermore, to discard the effect of the scale of function $\mathcal{L}_\epsilon(\theta)$ for different $\epsilon$, we estimate the scale of $\mathcal{L}_\epsilon(\theta)$ by randomly sampling $\theta$. We then normalize the top Hessian eigenvalues by the average value of $\mathcal{L}_\epsilon(\theta)$ on these random samples. In addition, we show the learning curve of $\mathcal{L}_\epsilon(\theta)$ on the training set during training in Figure 11 of Appendix C.2.2. It clearly show similar magnitude of $\mathcal{L}_\epsilon(\theta)$ for different values of $\epsilon$.

In Figure 3, we show the top 20 Hessian eigenvalues, both before and after normalization, of CIFAR10 models under different adversarial budgets. We also provide 3D visualizations of the neighborhood in the directions of the top 2 eigenvectors in Figure 12 of Appendix C.2.2. It is clear that the local effective second-order Lipschitz constant of the model obtained consistently increases with the value of $\epsilon$. That is, the minima found in $\mathcal{L}_\epsilon(\theta)$ are sharper under larger $\epsilon$.

To validate the claim in Section 3.2 that non-smoothness arises from abrupt changes of the adversarial examples, we study the similarity of adversarial perturbations generated by different model parameter values in a small neighborhood. Specifically, we perturb the model parameters $\theta$ in opposite directions

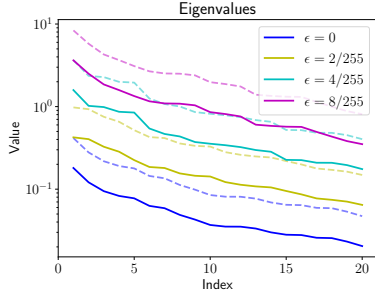
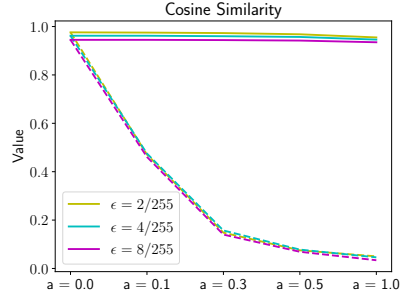

Figure 3: Top 20 eigenvalues of the Hessian matrix for ResNet18 models. We show the normalized (solid) and original (dashed) values.

Figure 4: Cosine similarity between perturbations $\mathbf{x}'_{a\mathbf{v}} - \mathbf{x}$ and $\mathbf{x}'_{-a\mathbf{v}} - \mathbf{x}$. $\mathbf{v}$ is either the top eigenvector (dashed) or random (solid).

to $\theta + a\mathbf{v}$ and $\theta - a\mathbf{v}$, where $\mathbf{v}$ is a unit vector and $a$ is a scalar. Let $\mathbf{x}'_{a\mathbf{v}}$ and $\mathbf{x}'_{-a\mathbf{v}}$ represent the adversarial examples generated by the corresponding model parameters. We then calculate the average cosine similarity between the perturbation $\mathbf{x}'_{a\mathbf{v}} - \mathbf{x}$ and $\mathbf{x}'_{-a\mathbf{v}} - \mathbf{x}$ over the training set.

The results on CIFAR10 models are provided in Figure 4. To account for the random start in PGD, we run each experiment $4$ times and report the average value. The variances of all experiments are smaller than $0.005$ and thus not shown in the figure. Note that, when $\mathbf{v}$ is a random unit vector, the robust error $\mathcal{E}_\epsilon(\theta)$ of the parameters $\theta \pm a\mathbf{v}$ on both the training and test sets remains unchanged for different values of $a$, indicating a flat landscape in the direction $\mathbf{v}$. The adversarial examples in this case are mostly similar and have very high cosine similarity. By contrast, if $\mathbf{v}$ is the top eigenvector of the Hessian matrix, i.e., the most curvy direction, then we see a sharp increase in the robust error $\mathcal{E}_\epsilon(\theta)$ when we increase $a$. Correspondingly, the cosine similarity between the adversarial perturbations is much lower, which indicates dramatic changes of the adversarial examples. We perform the same experiments on MNIST models in Appendix C.2.2 with the same observations.

## 5  Periodic Adversarial Scheduling

In Sections 3 and 4, we have theoretically and empirically shown that the adversarial loss landscape becomes less favorable to optimization under large adversarial budgets. In this section, we introduce a simple adversarial budget scheduling scheme to overcome these problems.

Inspired by the learning rate warmup heuristic used in deep learning [19, 25], we introduce warmup for the adversarial budget. Let $d$ be the current epoch index and $D$ be the warmup period's length. We define a cosine scheduler $\epsilon_{cos}$ and a linear scheduler $\epsilon_{lin}$, parameterized by $\epsilon_{max}$ and $\epsilon_{min}$, as

$$\epsilon_{cos}(d) = \frac{1}{2}(1 - \cos \frac{d}{D}\pi)(\epsilon_{max} - \epsilon_{min}) + \epsilon_{min}, \; \epsilon_{lin}(d) = (\epsilon_{max} - \epsilon_{min})\frac{d}{D} + \epsilon_{min} \;. \quad (5)$$

We clip $\epsilon_{cos}(d)$ and $\epsilon_{lin}(d)$ between $0$ and $\epsilon_{target}$, the target value of $\epsilon$. If $\epsilon_{min} \leq 0$ and $\epsilon_{max} > \epsilon_{target}$, the value of $\epsilon$ starts from $0$, gradually increases to $\epsilon_{target}$ and remains constant then.

This warmup strategy allows us to overcome the fact, highlighted in the previous sections, that adversarial training is more sensitive to the learning rate under a large budget because the gradients are more scattered. This is evidenced by Figure 5, which compares the robust test error of MNIST models relying on different adversarial budget scheduling schemes. For all models, we used $\epsilon = 0.4$, and report results after 100 epochs with different but constant learning rates in Adam [28]. Our linear and cosine schedulers perform better than using a constant value of $\epsilon$ during training and yield good performance for a broader range of learning rates: in the small learning rate regime, they speed up training; in the large learning rate regime, they stabilize training and avoid divergence. Note that, as shown in Appendix C.2.3, warmup of the learning rate does not yield similar benefits.

As shown in [25], periodic learning rates enable model ensembling to improve the performance. Here, we can follow the same strategy but also for the adversarial budget. To this end, we divide the training phase into several periods and store one model at the end of each period. We make final predictions based on the ensemble of these models. This periodic scheme has no computational overhead. We call it periodic adversarial scheduling (PAS).

As before, we run experiments on MNIST and CI-FAR10. For MNIST, we train each model for 100 epochs and do not use a periodic scheduling for the learning rate, which we found not to improve the results even if we use a constant adversarial budget. For CIFAR10, we train each model for 200 epochs. When there are no learning rate resets, our results indicate the final model after 200 epochs. When using a periodic learning rate, we divide the 200 epochs into 3 periods, i.e., we reset the learning rate and the adversarial budget after 100 and 150 epochs, and compute the results using an ensemble of these 3 models. The value of learning rate and the adversarial budget size are calculated based on the ratio of the current epoch index to the current period length. We provide more details about hyper-parameter settings in Appendix C.1.

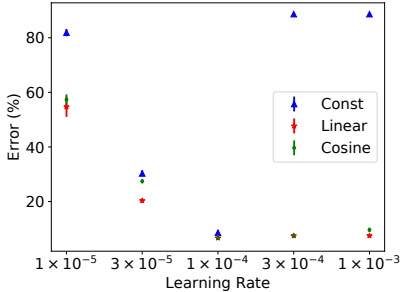

Figure 5: Mean and standard deviation of the test error under different learning rates with Adam and adversarial budget scheduling.

| Task | Periodic Learning Rate | $\epsilon$ Scheduler | Clean Error (%) | Robust Error (%) | | | | |
|---|---|---|---|---|---|---|---|---|
| | | | | PGD (%) | PGD100 (%) | APGD100 CE (%) | APGD100 DLR (%) | Square5K (%) |
| MNIST LeNet $\epsilon = 0.4$ | No | Constant | 1.56(17) | 8.58(89) | 10.86(143) | 15.18(155) | 14.70(136) | 19.58(45) |
| | | Cosine | 1.08(2) | 6.64(70) | 8.46(82) | 14.36(134) | 13.46(129) | 16.78(25) |
| | | Linear | 1.06(6) | 6.69(59) | 8.79(116) | 13.91(150) | 13.17(120) | 17.05(47) |
| CIFAR10 VGG $\epsilon = 8/255$ | No | Constant | 28.25(47) | 56.22(43) | 56.19(32) | 58.18(46) | 58.65(69) | 54.37(29) |
| | | Cosine | 25.06(19) | 56.06(48) | 56.00(42) | 57.83(45) | 58.88(16) | 53.95(15) |
| | | Linear | 23.56(95) | 56.09(14) | 55.88(5) | 57.74(16) | 58.39(18) | 53.66(24) |
| | Yes | Constant | 28.33(81) | 54.24(28) | 54.16(26) | 55.45(26) | 56.56(4) | 52.85(18) |
| | | Cosine | 23.91(21) | 53.18(21) | 53.10(18) | 54.44(16) | 55.80(24) | 51.41(37) |
| | | Linear | 21.88(33) | 53.03(14) | 52.97(17) | 54.32(17) | 55.63(17) | 51.28(4) |
| CIFAR10 ResNet18 $\epsilon = 8/255$ | No | Constant | 18.62(6) | 55.00(8) | 54.97(9) | 57.26(13) | 56.60(25) | 50.59(19) |
| | | Cosine | 18.43(26) | 53.95(23) | 53.85(21) | 56.16(18) | 55.77(24) | 49.60(18) |
| | | Linear | 18.55(14) | 53.46(20) | 53.41(10) | 55.69(17) | 55.45(22) | 49.66(28) |
| | Yes | Constant | 21.00(5) | 48.98(25) | 48.87(25) | 50.29(27) | 50.98(6) | 46.84(9) |
| | | Cosine | 19.90(18) | 48.57(25) | 48.49(27) | 49.71(22) | 50.54(9) | 46.19(11) |
| | | Linear | 20.26(28) | 48.60(13) | 48.52(13) | 49.73(9) | 50.68(11) | 46.47(26) |

Table 1: Comparison between different adversarial budget schedulers under different adversarial attacks. *Cosine / Linear schedulers* are consistently better than *constant schedulers*. The number between brackets indicate the standard deviation across different runs. Specifically, for example, $1.56(17)$ stands for $1.56 \pm 0.17$.

We compare different scheduler in adversarial budget under different tasks and settings. We evaluate the robustness of our trained models by different kinds of attacks. First we evaluate the models under the PGD attack used in training (PGD), i.e., 50-iteration PGD for MNIST models and 10-iteration PGD for CIFAR10 models. Then, we increase the number of iterations in PGD and compute the robust error under 100-iteration PGD. To solve the issue of suboptimal step size, we also evaluate our models using the state-of-the-art AutoPGD attack [10], which search for the optimal step sizes. We run AutoPGD for 100 iterations for evaluation, based on either cross-entropy loss (APGD100 CE) or the difference of logit ratio loss (APGD100 DLR). To avoid gradient masking, we also run the state-of-the-art black-box SquareAttack [3] for 5000 iterations (Square5K). The hyperparameter details are defered in Appendix C.1.

The results are summarized in Table 1, where we compare the clean and robust accuracy under different adversarial attacks on the test set. It is clear that our proposed cosine or linear schedulers yield better performance, in both clean accuracy and robust accuracy, than using a constant adversarial budget in all cases. For MNIST, warmup not only makes training robust to different choices of learning rate, but also improves the final robust accuracy. For CIFAR10, model ensembling enabled by the periodic scheduler improves the robust accuracy.

## 6   Discussion

**Model capacity.**   In addition to the size of the adversarial budget, the capacity of the model also greatly affects the adversarial loss landscape and thus the performance of adversarial training.

Adversarial training needs higher model capacity in two aspects: if we decrease the model capacity, adversarial training will fail to converge while vanilla training still works [33]; if we increase the model capacity, the robust accuracy of adversarial training continues to rise while the clean accuracy of normal training saturates [50]. Furthermore, we show in Appendix C.2.4 that smaller models are more likely to have dead layers because of their lower dimensionality. As a result, warmup in adversarial budget is also necessary for small models. In many cases, the parameter space of small models has good minima in terms of robustness, but adversarial training with a constant value of $\epsilon$ fails to find them. For example, one can obtain small but robust models by pruning large ones [21, 52].

**Architecture.** The network architecture encodes the parameterization of the model, so it greatly affects the adversarial loss landscape. For example, in Table 1, ResNet18 has fewer trainable parameters but better performance than VGG on CIFAR10, indicating that ResNet18 has a better parameterization in terms of robustness. Since the optimal architecture for adversarial robustness is not necessarily the same as the one for clean accuracy, we believe that finding architectures inherently favorable to adversarial training is an interesting but challenging topic for future research.

**Connectivity of minima.** Local minima in the vanilla loss landscape are well-connected [12, 14]: there exist flat hyper curves connecting them. In Appendix C.2.5, we study the connectivity of converged model parameters in the adversarial setting. We find that the parameters of two adversarially trained models are less connected in the adversarial loss landscape than in the vanilla setting. That is, the path connecting them needs to go over suboptimal regions.

**Adversarial example generation** We approximate the adversarial loss using adversarial examples generated by PGD, which is a good estimate of the inner maximization in (1). PGD-based adversarial training updates model parameters by near-optimal adversarial examples. However, recent works [41, 47] have shown that robust models can also be trained by suboptimal adversarial examples, which are faster to obtain. The formulation of these methods differs from (1), because the inner maximization problem is not approximately solved. Understanding why models (partially) trained on suboptimal adversarial examples are resistant to stronger adversarial examples needs more investigation.

# 7    Conclusion

We have studied the properties of the loss landscape under adversarial training. We have shown that the adversarial loss landscape is non-smooth and not favorable to optimization, due to the dependency of adversarial examples on the model parameters. Furthermore, we have empirically evidenced that large adversarial budgets slow down training in the early stages and impedes convergence in the end. Finally, we have demonstrated the advantages of warmup and periodic scheduling of the adversarial budget size during training. They make training more robust to different choices of learning rate and yield better performance than vanilla adversarial training.

# 8    Broader Impact

The existence of adversarial examples has raised serious concerns about the deployment of deep learning models in safety-sensitive domains, such as medical imaging [32] and autonomous navigation [1]. In these domains, as in many others, adversarial training remains the most popular, effective, and general method to train robust models. By studying the nature of optimization in adversarial training and proposing solutions to overcome the underlying challenges, our work has potential for high societal impact in these fields. Although the robust accuracy is much lower than the clean accuracy so far, the intrinsic properties of adversarial training we have discovered open up future research directions to improve its performance. From an ecological perspective, however, we acknowledge that the higher computational cost of adversarial training translates to higher carbon footprint than vanilla training. Nevertheless, we believe that the potential societal benefits of robustness to attacks outweigh this drawback.

# 9    Acknowledgements

We thankfully acknowledge the support of the Hasler Foundation (Grant No. 16076) for this work.

## Footnotes

[1] Strictly speaking, $\mathcal{L}_\epsilon(\theta)$ is not differentiable at some point, so $\nabla_\theta \mathcal{L}_\epsilon(\theta)$ might be ill-defined. In this paper, we use $\nabla_\theta \mathcal{L}_\epsilon(\theta)$ for simplicity. Nevertheless, the inequality holds for any subgradient $\mathbf{v} \in \partial_\theta \mathcal{L}_\epsilon(\theta)$.

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
