[Supplementary Material]

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

# A Theoretical Analysis

## A.1 Binary Logistic Regression

In this section, we discuss binary logistic regression. In this case, $K = 2$ and the logit function is $f(\mathbf{w}) = \left[\mathbf{w}^T\mathbf{x}, -\mathbf{w}^T\mathbf{x}\right]$, where $\mathbf{w} \in \mathbb{R}^m$ is the only trainable parameter. If we use $+1$ and $-1$ to label both classes, then the overall loss function for a dataset $\{(\mathbf{x}_i, y_i)\}_{i=1}^N$ is $\mathcal{L}(\mathbf{w}) = \frac{1}{N}\sum_{i=1}^N \log\left(1 + e^{-y_i\mathbf{w}^T\mathbf{x}_i}\right)$. Under the adversarial budget $\mathcal{S}_\epsilon^{(p)}(\mathbf{x})$, the corresponding adversarial loss function is $\mathcal{L}_\epsilon(\mathbf{w}) = \frac{1}{N}\sum_{i=1}^N \log\left(1 + e^{-y_i\mathbf{w}^T\mathbf{x}_i + \epsilon\|\mathbf{w}\|_q}\right)$, where $l_q$ is the dual norm of $l_p$. Since the magnitude of $\mathbf{w}$ does not change the results of the classifier, we can assume $\|\mathbf{w}\|_q = 1$ without loss of generality. As a result, the adversarial loss function is

$$\mathcal{L}_\epsilon(\mathbf{w}) = \frac{1}{N}\sum_{i=1}^N \log\left(1 + e^{-y_i\mathbf{w}^T\mathbf{x}_i + \epsilon}\right). \tag{6}$$

The following theorem describes the properties of $\mathcal{L}_\epsilon(\mathbf{w})$ for different values of $\epsilon$.

**Theorem 3.** *If the dataset $\{(\mathbf{x}_i, y_i)\}_{i=1}^N$ is linearly separable under the adversarial budget $\mathcal{S}_{\widehat{\epsilon}}(\mathbf{x})$, then for any unit vector $\mathbf{m} \in \mathbb{R}^m$ and values $\epsilon_1$, $\epsilon_2$ such that $\epsilon_1 \leq \epsilon_2 \leq \widehat{\epsilon}$, we have $\mathbf{m}^T\nabla_\mathbf{w}^2\mathcal{L}_{\epsilon_1}(\mathbf{w})\mathbf{m} \leq \mathbf{m}^T\nabla_\mathbf{w}^2\mathcal{L}_{\epsilon_2}(\mathbf{w})\mathbf{m}$. More specifically, both the largest and smallest eigenvalue of $\nabla_\mathbf{w}^2\mathcal{L}_{\epsilon_1}(\mathbf{w})$ are no greater than those of $\nabla_\mathbf{w}^2\mathcal{L}_{\epsilon_2}(\mathbf{w})$.*

We provide the proof of Theorem 3 in Appendix B.5. Since $\mathbf{m}^T\nabla_\mathbf{w}^2\mathcal{L}_\epsilon(\mathbf{w})\mathbf{m}$ is the curvature of $\mathcal{L}_\epsilon(\mathbf{w})$ in the direction of $\mathbf{m}$, Theorem 3 shows that the curvature of $\mathcal{L}_\epsilon(\mathbf{w})$ increases with $\epsilon$ in any direction if the whole dataset is linearly separable. For an individual data point $\mathbf{x}$, if $\forall \mathbf{x}' \in \mathcal{S}_{\widehat{\epsilon}}(\mathbf{x})$, $\mathbf{x}'$ is correctly classified, then the curvature of $g_\epsilon(\mathbf{x}, \mathbf{w})$ also increases with $\epsilon$ in any direction as long as $\epsilon \leq \widehat{\epsilon}$. The assumption for an individual point here is much weaker than the one in Theorem 3. If the overwhelming majority of the data points are correctly classified under the adversarial budget $\mathcal{S}_{\widehat{\epsilon}}$, the conclusion still holds in practice.

## A.2 Discussions of ReLU Networks

Unlike sigmoid or tanh, ReLU is not a smooth function. However, it is smooth *almost everywhere*, except at $0$. As a result, we can make the following assumptions for the function $g$ represented by a ReLU network.

**Assumption 2.** *The function $g$ satisfies the following conditions:*

$$\begin{aligned}
\|g(\mathbf{x}, \theta_1) - g(\mathbf{x}, \theta_2)\| &\leq L_\theta\|\theta_1 - \theta_2\|, \\
\|\nabla_\theta g(\mathbf{x}, \theta_1) - \nabla_\theta g(\mathbf{x}, \theta_2)\| &\leq L_{\theta\theta}\|\theta_1 - \theta_2\| + D_{\theta\theta}, \\
\|\nabla_\theta g(\mathbf{x}_1, \theta) - \nabla_\theta g(\mathbf{x}_2, \theta)\| &\leq L_{\theta\mathbf{x}}\|\mathbf{x}_1 - \mathbf{x}_2\|_p + D_{\theta\mathbf{x}}.
\end{aligned} \tag{7}$$

We adjust the second-order smoothness assumption by adding two constants $D_{\theta\theta}$, $D_{\theta\mathbf{x}}$. They are the upper bound of the gradient difference in the neighborhood of non-smooth points. Therefore, they measures how abruptly the (sub)gradients can change in a sufficiently small region in the parameter space and can be considered as a quantitative measure of *gradient scattering*.

The following corollary states the properties of $g_\epsilon$ under Assumption 2.

**Corollary 1.** *If Assumption 2 is satisfied, then we have*

$$\begin{aligned}
\|\mathcal{L}_\epsilon(\theta_1) - \mathcal{L}_\epsilon(\theta_2)\| &\leq L_\theta\|\theta_1 - \theta_2\| \\
\|\nabla_\theta\mathcal{L}_\epsilon(\theta_1) - \nabla_\theta\mathcal{L}_\epsilon(\theta_2)\| &\leq L_{\theta\theta}\|\theta_1 - \theta_2\| + 2\epsilon L_{\theta\mathbf{x}} + D_{\theta\theta} + D_{\theta\mathbf{x}}.
\end{aligned} \tag{8}$$

The proof directly follows the one of Proposition 2. As in Proposition 2, the additional $2\epsilon L_{\theta\mathbf{x}}$ term in Corollary 1 evidences more severe *gradient scattering* under adversarial training in the context of ReLU networks, which harms optimization.

Similarly, we can easily extend the study of the asymptotic gradient magnitude of Theorem 2 to the account for Assumption 2.

**Corollary 2.** *Let Assumption 2 hold, the stochastic gradient $\nabla_\theta \widehat{\mathcal{L}}_\epsilon(\theta_t)$ be unbiased and have bounded variance, and the SGD update $\theta_{t+1} = \theta_t - \alpha_t \nabla_\theta \widehat{\mathcal{L}}_\epsilon(\theta_t)$ use a constant step size $\alpha_t = \alpha = \frac{1}{L_{\theta\theta}\sqrt{T}}$ for $T$ iterations. Given the trajectory of the parameters during optimization $\{\theta_t\}_{t=1}^T$, then we can bound the asymptotic probability of large gradients for a sufficiently large $T$ as*

$$\forall \gamma \geq 2, P(\|\nabla_\theta \mathcal{L}_\epsilon(\theta_t)\| > \gamma(\epsilon L_{\theta\mathbf{x}} + \frac{1}{2}D_{\theta\theta} + \frac{1}{2}D_{\theta\mathbf{x}})) < \frac{4}{\gamma^2 - 2\gamma + 4} . \tag{9}$$

## B Proofs

### B.1 Proof of Proposition 1

*Proof.* For arbitrary $\mathbf{W} \in \mathcal{V}_{\epsilon_2}$, we have $\forall i \in [K], \mathbf{x}' \in \mathcal{S}_{\epsilon_2}(\mathbf{x}), (\mathbf{w}_i - \mathbf{w}_y)\mathbf{x}' \leq 0$ based on the definition of $\mathcal{V}_\epsilon$.

Since $\epsilon_1 \leq \epsilon_2$, we have $\mathcal{S}_{\epsilon_1}(\mathbf{x}) \subseteq \mathcal{S}_{\epsilon_2}(\mathbf{x})$. As a result, $\forall i \in [K], \mathbf{x}' \in \mathcal{S}_{\epsilon_1}(\mathbf{x}), (\mathbf{w}_i - \mathbf{w}_y)\mathbf{x}' \leq 0$. That is to say, $\mathbf{W} \in \mathcal{V}_{\epsilon_1}$. $\mathbf{W}$ is arbitrarily picked, so $\mathcal{V}_{\epsilon_2} \subseteq \mathcal{V}_{\epsilon_1}$.

$\square$

### B.2 Proof of Theorem 1

*Proof.* In multi-class logistic regression, as discussed in Section 3.1, the function $g(\mathbf{x}, \mathbf{W}) = \log\left(1 + \sum_{j \neq y} \exp^{(\mathbf{w}_j - \mathbf{w}_y)\mathbf{x}}\right)$ is a convex function w.r.t. the parameters $\mathbf{W}$, and so is $g_\epsilon(\mathbf{x}, \mathbf{W})$. Based on convexity, for any $\mathbf{W} \in \mathcal{T}_\epsilon$, the statement $0 \in \arg\min_\gamma g_\epsilon(\mathbf{x}, \gamma\mathbf{W})$ is equivalent to the following statement:

$$\forall \Delta\gamma > 0, g_\epsilon(\mathbf{x}, \Delta\gamma\mathbf{W}) \geq g_\epsilon(\mathbf{x}, \mathbf{0}) \text{ and } g_\epsilon(\mathbf{x}, -\Delta\gamma\mathbf{W}) \geq g_\epsilon(\mathbf{x}, \mathbf{0}) . \tag{10}$$

Note that $g_\epsilon(\mathbf{x}, \mathbf{0}) \equiv \log K$, which means that the loss of the model is independent of both the input and the adversarial budget when $\mathbf{W} = \mathbf{0}$. Given $\epsilon_1 \geq \epsilon_2$, we have, $\forall \mathbf{x}, \mathbf{W}, g_{\epsilon_1}(\mathbf{x}, \mathbf{W}) \geq g_{\epsilon_2}(\mathbf{x}, \mathbf{W})$. Therefore, for an arbitrary $\mathbf{W} \in \mathcal{T}_{\epsilon_2}$, we have the following inequality:

$$\forall \Delta\gamma > 0, g_{\epsilon_1}(\mathbf{x}, \Delta\gamma\mathbf{W}) \geq g_{\epsilon_2}(\mathbf{x}, \Delta\gamma\mathbf{W}) \geq g_{\epsilon_2}(\mathbf{x}, \mathbf{0}) = g_{\epsilon_1}(\mathbf{x}, \mathbf{0}) . \tag{11}$$

The first inequality is based on $\epsilon_1 \geq \epsilon_2$, the second one is based on (10) and the last one arises from the fact that, $\forall\epsilon, g_\epsilon(\mathbf{x}, \mathbf{0})$ is a constant. Similarly, we also have $g_{\epsilon_1}(\mathbf{x}, -\Delta\gamma\mathbf{W}) \geq g_{\epsilon_1}(\mathbf{x}, \mathbf{0})$. Therefore, we have $\mathbf{W} \in \mathcal{T}_{\epsilon_1}$, which means $\mathcal{T}_{\epsilon_2} \subseteq \mathcal{T}_{\epsilon_1}$.

To prove the second half of Theorem 1, one barrier is that we do not have an analytical form for $g_\epsilon(\mathbf{x}, \mathbf{W})$. Instead, we introduce a lower bound $\underline{g_\epsilon}(\mathbf{x}, \mathbf{W})$ of $g_\epsilon(\mathbf{x}, \mathbf{W})$, which has an analytical form.

We consider the perturbation $\mathbf{x}' = \mathbf{x} + \epsilon \frac{(\mathbf{w}_m - \mathbf{w}_y)^{\frac{q}{p}}}{\|\mathbf{w}_m - \mathbf{w}_y\|_q^{\frac{q}{p}}}$ [2], where $m = \arg\max_j \|\mathbf{w}_j - \mathbf{w}_y\|_q$. It can be verified that $\mathbf{x}' \in \mathcal{S}_\epsilon^{(p)}(\mathbf{x})$. Therefore, we set $\underline{g_\epsilon}(\mathbf{x}, \mathbf{W}) = g(\mathbf{x}', \mathbf{W})$, which is a valid lower bound of $g_\epsilon(\mathbf{x}, \mathbf{W})$. Then, the analytical expression of $\underline{g_\epsilon}(\mathbf{x}, \mathbf{W})$ can be written as

$$\underline{g_\epsilon}(\mathbf{x}, \mathbf{W}) = \log\left(1 + \exp^{(\mathbf{w}_m - \mathbf{w}_y)\mathbf{x} + \epsilon\|\mathbf{w}_m - \mathbf{w}_y\|_q} + \sum_{j \neq y, j \neq m} \exp^{(\mathbf{w}_j - \mathbf{w}_y)\mathbf{x} + \epsilon(\mathbf{w}_j - \mathbf{w}_y)\frac{(\mathbf{w}_m - \mathbf{w}_y)^{\frac{q}{p}}}{\|\mathbf{w}_m - \mathbf{w}_y\|_q^{\frac{q}{p}}}}\right) . \tag{12}$$

Since $m = \arg\max_j \|\mathbf{w}_j - \mathbf{w}_y\|_q$, then $(\mathbf{w}_j - \mathbf{w}_y)\frac{(\mathbf{w}_m - \mathbf{w}_y)^{\frac{q}{p}}}{\|\mathbf{w}_m - \mathbf{w}_y\|_q^{\frac{q}{p}}} \leq \|\mathbf{w}_m - \mathbf{w}_y\|_q$. As a result, if $\epsilon$ is large enough, the second term inside the logarithm of (12) will dominate the summation and

$\lim_{\epsilon \to \infty} \underline{g}_\epsilon(\mathbf{x}, \mathbf{W}) = \infty$. More specifically, we can find $\bar{\epsilon} = \frac{\log(K-1) - (\mathbf{w}_m - \mathbf{w}_y)\mathbf{x}}{\|\mathbf{w}_m - \mathbf{w}_y\|_q}$, such that, $\forall \epsilon > \bar{\epsilon}$, $\mathbf{W}$, then $\underline{g}_\epsilon(\mathbf{x}, \mathbf{W}) \geq \log K = g_\epsilon(\mathbf{x}, \mathbf{0})$.

Now, $\forall \mathbf{W} \in \mathbb{R}^{m \times K}, \Delta\gamma > 0, \epsilon \geq \bar{\epsilon}$, we have $g_\epsilon(\mathbf{x}, \Delta\gamma\mathbf{W}) \geq \underline{g}_\epsilon(\mathbf{x}, \Delta\gamma\mathbf{W}) \geq g_\epsilon(\mathbf{x}, \mathbf{0})$. Similarly, we have $g_\epsilon(\mathbf{x}, -\Delta\gamma\mathbf{W}) \geq g_\epsilon(\mathbf{x}, \mathbf{0})$. As a result, we have, $\forall \mathbf{W} \in \mathbb{R}^{m \times K}, g_\epsilon(\mathbf{x}, \mathbf{W}) \geq g_\epsilon(\mathbf{x}, \mathbf{0})$, so $\mathbf{0} \in \arg\min_{\mathbf{W}} g_\epsilon(\mathbf{x}, \mathbf{W})$. Based on (10), we have $\mathcal{T}_\epsilon = \mathbb{R}^{m \times K}$.

$\square$

### B.3 Proof of Proposition 2

*Proof.* Recall that $\mathcal{L}_\epsilon(\theta)$ is the average of $g_\epsilon(\mathbf{x}, \theta)$ over the dataset. Therefore, to prove Proposition 2, we only need to prove the following inequalities for any data point $\mathbf{x}$:

$$\|g_\epsilon(\mathbf{x}, \theta_1) - g_\epsilon(\mathbf{x}, \theta_2)\| \leq L_\theta \|\theta_1 - \theta_2\| \,,$$
$$\|\nabla_\theta g_\epsilon(\mathbf{x}, \theta_1) - \nabla_\theta g_\epsilon(\mathbf{x}, \theta_2)\| \leq L_{\theta\theta}\|\theta_1 - \theta_2\| + 2\epsilon L_{\theta\mathbf{x}} \,. \quad (13)$$

To prove the first inequality, we introduce the adversarial examples for parameter $\theta_1$ and $\theta_2$:

$$\mathbf{x}_1 = \arg\max_{\mathbf{x}' \in \mathcal{S}_\epsilon^{(p)}(\mathbf{x})} g(\mathbf{x}', \theta_1) \,,$$
$$\mathbf{x}_2 = \arg\max_{\mathbf{x}' \in \mathcal{S}_\epsilon^{(p)}(\mathbf{x})} g(\mathbf{x}', \theta_2) \,. \quad (14)$$

Therefore, $g_\epsilon(\mathbf{x}, \theta_1) = g(\mathbf{x}_1, \theta_1)$ and $g_\epsilon(\mathbf{x}, \theta_2) = g(\mathbf{x}_2, \theta_2)$.

By definition, we have $g(\mathbf{x}_1, \theta_1) \geq g(\mathbf{x}_2, \theta_1)$ and $g(\mathbf{x}_2, \theta_2) \geq g(\mathbf{x}_1, \theta_2)$. As a result, $\|g_\epsilon(\mathbf{x}, \theta_1) - g_\epsilon(\mathbf{x}, \theta_2)\| = \|g(\mathbf{x}_1, \theta_1) - g(\mathbf{x}_2, \theta_2)\|$. If $g(\mathbf{x}_1, \theta_1) - g(\mathbf{x}_2, \theta_2) \leq 0$, we have

$$\|g_\epsilon(\mathbf{x}, \theta_1) - g_\epsilon(\mathbf{x}, \theta_2)\| = g(\mathbf{x}_2, \theta_2) - g(\mathbf{x}_1, \theta_1) \leq g(\mathbf{x}_2, \theta_2) - g(\mathbf{x}_2, \theta_1) \leq L_\theta\|\theta_1 - \theta_2\| \,. \quad (15)$$

Similarly, if $g(\mathbf{x}_1, \theta_1) - g(\mathbf{x}_2, \theta_2) \geq 0$, we have

$$\|g_\epsilon(\mathbf{x}, \theta_1) - g_\epsilon(\mathbf{x}, \theta_2)\| = g(\mathbf{x}_1, \theta_1) - g(\mathbf{x}_2, \theta_2) \leq g(\mathbf{x}_1, \theta_1) - g(\mathbf{x}_1, \theta_2) \leq L_\theta\|\theta_1 - \theta_2\| \,. \quad (16)$$

This proves the first inequality in (13). The bound is tight, and equality is achieved when, for example, $\mathbf{x}_1 = \mathbf{x}_2$.

The second inequality in (13) is more straightforward. We have

$$\begin{aligned}
\|\nabla_\theta g_\epsilon(\mathbf{x}, \theta_1) - \nabla_\theta g_\epsilon(\mathbf{x}, \theta_2)\| &= \|\nabla_\theta g(\mathbf{x}_1, \theta_1) - \nabla_\theta g(\mathbf{x}_2, \theta_2)\| \\
&= \|\nabla_\theta g(\mathbf{x}_1, \theta_1) - \nabla_\theta g(\mathbf{x}_1, \theta_2) + \nabla_\theta g(\mathbf{x}_1, \theta_2) - \nabla_\theta g(\mathbf{x}_2, \theta_2)\| \\
&\leq \|\nabla_\theta g(\mathbf{x}_1, \theta_1) - \nabla_\theta g(\mathbf{x}_1, \theta_2)\| + \|\nabla_\theta g(\mathbf{x}_1, \theta_2) - \nabla_\theta g(\mathbf{x}_2, \theta_2)\| \\
&\leq L_{\theta\theta}\|\theta_1 - \theta_2\| + L_{\theta\mathbf{x}}\|\mathbf{x}_1 - \mathbf{x}_2\|_p \\
&\leq L_{\theta\theta}\|\theta_1 - \theta_2\| + 2\epsilon L_{\theta\mathbf{x}} \,.
\end{aligned} \quad (17)$$

The last inequality in (17) is satisfied because both $\mathbf{x}_1$ and $\mathbf{x}_2$ belong to $\mathcal{S}_\epsilon^{(p)}(\mathbf{x})$. This bound is tight, and equality is reached only when $\|\mathbf{x}_1 - \mathbf{x}_2\|_p = 2\epsilon$.

$\square$

### B.4 Proof of Theorem 2

*Proof.* Let $\sigma^2$ to denote the variance of stochastic gradient $\nabla_\theta \widehat{\mathcal{L}}_\epsilon(\theta)$. Based on the assumption that $\nabla_\theta \widehat{\mathcal{L}}_\epsilon(\theta)$ is unbiased, we have

$$\mathbb{E}[\nabla_\theta \widehat{\mathcal{L}}_\epsilon(\theta)] = \nabla_\theta \mathcal{L}_\epsilon(\theta) \,,$$
$$\mathbb{E}\|\nabla_\theta \widehat{\mathcal{L}}_\epsilon(\theta)\|^2 = \|\nabla_\theta \mathcal{L}_\epsilon(\theta)\|^2 + \sigma^2 \,. \quad (18)$$

Proposition 2 shows that $\mathcal{L}_\epsilon(\theta)$ is continuous. Therefore, we introduce $\tilde{\theta}_t(u) = \theta_t + u(\theta_{t+1} - \theta_t)$ and derive an upper bound of $\mathcal{L}_\epsilon(\theta_{t+1}) - \mathcal{L}_\epsilon(\theta_t)$ by first order Taylor expansion and using the update

rule $\theta_{t+1} = \theta_t - \alpha_t \nabla_\theta \widehat{\mathcal{L}}_\epsilon(\theta_t)$. This yields

$$
\begin{aligned}
\mathcal{L}_\epsilon(\theta_{t+1}) - \mathcal{L}_\epsilon(\theta_t) &= \int_0^1 \langle \theta_{t+1} - \theta_t, \nabla_\theta \mathcal{L}_\epsilon(\tilde{\theta}_t(u)) \rangle d_u \\
&= \int_0^1 \langle -\alpha_t \nabla_\theta \widehat{\mathcal{L}}_\epsilon(\theta_t), \nabla_\theta \mathcal{L}_\epsilon(\tilde{\theta}_t(u)) \rangle d_u \\
&= \int_0^1 \langle -\alpha_t \nabla_\theta \widehat{\mathcal{L}}_\epsilon(\theta_t), \nabla_\theta \mathcal{L}_\epsilon(\tilde{\theta}_t(u)) - \nabla_\theta \mathcal{L}_\epsilon(\theta_t) \rangle d_u + \langle -\alpha_t \nabla_\theta \widehat{\mathcal{L}}_\epsilon(\theta_t), \nabla_\theta \mathcal{L}_\epsilon(\theta_t) \rangle \\
&\leq \int_0^1 \alpha_t \|\nabla_\theta \widehat{\mathcal{L}}_\epsilon(\theta_t)\| \|\nabla_\theta \mathcal{L}_\epsilon(\tilde{\theta}_t(u)) - \nabla_\theta \mathcal{L}_\epsilon(\theta_t)\| d_u - \alpha_t \langle \nabla_\theta \widehat{\mathcal{L}}_\epsilon(\theta_t), \nabla_\theta \mathcal{L}_\epsilon(\theta_t) \rangle \\
&\leq \int_0^1 \alpha_t \|\nabla_\theta \widehat{\mathcal{L}}_\epsilon(\theta_t)\| (L_{\theta\theta} \|\tilde{\theta}_t(u) - \theta_t\| + 2\epsilon L_{\theta\mathbf{x}}) d_u - \alpha_t \langle \nabla_\theta \widehat{\mathcal{L}}_\epsilon(\theta_t), \nabla_\theta \mathcal{L}_\epsilon(\theta_t) \rangle \\
&= \frac{1}{2} \alpha_t^2 L_{\theta\theta} \|\nabla_\theta \widehat{\mathcal{L}}_\epsilon(\theta_t)\|^2 + 2\epsilon L_{\epsilon\mathbf{x}} \alpha_t \|\nabla_\theta \widehat{\mathcal{L}}_\epsilon(\theta_t)\| - \alpha_t \langle \nabla_\theta \widehat{\mathcal{L}}_\epsilon(\theta_t), \nabla_\theta \mathcal{L}_\epsilon(\theta_t) \rangle .
\end{aligned}
\tag{19}
$$

Here, the first inequality comes from Hölder's Inequality; the second one follows the conclusion of Proposition 2.

By taking the expectation over the noise introduced by SGD, we have

$$
\begin{aligned}
\mathbb{E}[\mathcal{L}_\epsilon(\theta_{t+1})] - \mathbb{E}[\mathcal{L}_\epsilon(\theta_t)] &\leq \frac{1}{2} \alpha_t^2 L_{\theta\theta} (\|\nabla_\theta \mathcal{L}_\epsilon(\theta_t)\|^2 + \sigma^2) + 2\epsilon L_{\theta\mathbf{x}} \alpha_t \|\nabla_\theta \mathcal{L}_\epsilon(\theta_t)\| - \alpha_t \|\nabla_\theta \mathcal{L}_\epsilon(\theta_t)\|^2 \\
&= (\frac{1}{2} \alpha_t^2 L_{\theta\theta} - \alpha_t) \|\nabla_\theta \mathcal{L}_\epsilon(\theta_t)\|^2 + 2\epsilon L_{\theta\mathbf{x}} \alpha_t \|\nabla_\theta \mathcal{L}_\epsilon(\theta_t)\| + \frac{1}{2} \alpha_t^2 \sigma^2 L_{\theta\theta} \\
&\leq -\frac{1}{2} \alpha_t \|\nabla_\theta \mathcal{L}_\epsilon(\theta_t)\|^2 + 2\epsilon L_{\theta\mathbf{x}} \alpha_t \|\nabla_\theta \mathcal{L}_\epsilon(\theta_t)\| + \frac{1}{2} \alpha_t^2 \sigma^2 L_{\theta\theta} .
\end{aligned}
\tag{20}
$$

We use the approximation $\mathbb{E}\|\nabla_\theta \widehat{\mathcal{L}}_\epsilon(\theta)\| \simeq \|\nabla_\theta \mathcal{L}_\epsilon(\theta)\|$ because the variance arises mainly from the term $\|\nabla_\theta \widehat{\mathcal{L}}_\epsilon(\theta_t)\|^2$. The last inequality is based on the fact that $\alpha_t = \alpha = \frac{1}{L_{\theta\theta}\sqrt{T}}$, so $\alpha_t L_{\theta\theta} = \frac{1}{\sqrt{T}} \leq 1$.

Let us now sum (20) over $t \in [T]$. This gives

$$
\begin{aligned}
\sum_{t=0}^T \left[ \frac{1}{2} \alpha_t \|\nabla_\theta \mathcal{L}_\epsilon(\theta_t)\|^2 - 2\epsilon L_{\theta\mathbf{x}} \alpha_t \|\nabla_\theta \mathcal{L}_\epsilon(\theta_t)\| \right] &\leq \mathcal{L}_\epsilon(\theta_0) - \mathbb{E}[\mathcal{L}_\epsilon(\theta_T)] + \frac{T}{2} \alpha_t^2 \sigma^2 L_{\theta\theta} \\
&\leq \mathcal{L}_\epsilon(\theta_0) - \mathcal{L}_\epsilon(\theta^*) + \frac{T}{2} \alpha_t^2 \sigma^2 L_{\theta\theta} .
\end{aligned}
\tag{21}
$$

We use $\theta^*$ to denote the global minimum since $\mathcal{L}_\epsilon(\theta)$ is lower bounded. By introducing $\alpha_t = \alpha = \frac{1}{L_{\theta\theta}\sqrt{T}}$ into the formulation, we obtain

$$
\begin{aligned}
\frac{1}{T} \sum_{t=0}^T \left[ \frac{1}{2} \|\nabla_\theta \mathcal{L}_\epsilon(\theta_t)\|^2 - 2\epsilon L_{\theta\mathbf{x}} \|\nabla_\theta \mathcal{L}_\epsilon(\theta_t)\| \right] &\leq \frac{1}{\alpha T} [\mathcal{L}_\epsilon(\theta_0) - \mathcal{L}_\epsilon(\theta^*)] + \frac{1}{2} \alpha \sigma^2 L_{\theta\theta} \\
&= \frac{1}{\sqrt{T}} \left[ L_{\theta\theta}(\mathcal{L}_\epsilon(\theta_0) - \mathcal{L}_\epsilon(\theta^*)) + \frac{1}{2} \sigma^2 \right] .
\end{aligned}
\tag{22}
$$

Since the righthand side of (22) converges to 0 as $T \to +\infty$, we have

$$
\lim_{T \to +\infty} \frac{1}{T} \sum_{t=0}^T \left[ \frac{1}{2} \|\nabla_\theta \mathcal{L}_\epsilon(\theta_t)\|^2 - 2\epsilon L_{\theta\mathbf{x}} \|\nabla_\theta \mathcal{L}_\epsilon(\theta_t)\| \right] \leq 0 .
\tag{23}
$$

Let us define $h(\theta_t) = \frac{1}{2} \|\nabla_\theta \mathcal{L}_\epsilon(\theta_t)\|^2 - 2\epsilon L_{\theta\mathbf{x}} \|\nabla_\theta \mathcal{L}_\epsilon(\theta_t)\|$ for notation simplicity. Then, inequality (23) shows that $\mathbb{E}_t[h(\theta_t)] \leq 0$ when $T$ is large enough.

We define $\widehat{\gamma} := \|\nabla_\theta \mathcal{L}_\epsilon(\theta_t)\| / (\epsilon L_{\theta\mathbf{x}})$, then we have $h(\theta_t) = (\frac{1}{2}\widehat{\gamma}^2 - 2\widehat{\gamma})\epsilon^2 L_{\theta\mathbf{x}}^2$. $h(\theta)$ is monotonically increasing when $\|\nabla_\theta \mathcal{L}_\epsilon(\theta_t)\| \geq 2\epsilon L_{\theta\mathbf{x}}$, so when $\widehat{\gamma} \geq 2$, $h(\theta_t) \geq (\frac{1}{2}\widehat{\gamma}^2 - 2\widehat{\gamma})\epsilon^2 L_{\theta\mathbf{x}}^2$. Considering

$h(\theta_t) \geq -2\epsilon^2 L_{\theta\mathbf{x}}^2$ always holds, we can then bound the probability of $P(\|\nabla_\theta \mathcal{L}_\epsilon(\theta_t)\| > \gamma\epsilon L_{\theta\mathbf{x}})$ when $\gamma > 2$ as follows:

$$\mathbb{E}_t[h(\theta_t)] > -2\epsilon^2 L_{\theta\mathbf{x}}^2(1 - P(\|\nabla_\theta \mathcal{L}_\epsilon(\theta_t)\| > \gamma\epsilon L_{\theta\mathbf{x}})) + (\frac{1}{2}\gamma^2 - 2\gamma)\epsilon^2 L_{\theta\mathbf{x}}^2 P(\|\nabla_\theta \mathcal{L}_\epsilon(\theta_t)\| > \gamma\epsilon L_{\theta\mathbf{x}}) .$$
(24)

Finally, by rearranging (24) and using $\mathbb{E}_t[h(\theta_t)] \leq 0$, we obtain

$$\forall \gamma > 2, \; P(\|\nabla_\theta \mathcal{L}_\epsilon(\theta_t)\| > \gamma\epsilon L_{\theta\mathbf{x}}) < \frac{4}{\gamma^2 - 2\gamma + 4} .$$
(25)

$\square$

## B.5 Proof of Theorem 3

To prove Theorem 3, let us first introduce the following lemma.

**Lemma 1.** *Given a vector set $\{\mathbf{x}_i\}_{i=1}^N$ and scalar sets $\{a_i\}_{i=1}^N$, $\{b_i\}_{i=1}^N$, we define $\mathbf{A} = \sum_{i=1}^N a_i \mathbf{x}_i \mathbf{x}_i^T$ and $\mathbf{B} = \sum_{i=1}^N b_i \mathbf{x}_i \mathbf{x}_i^T$. If, $\forall i, \; a_i \geq b_i$, then $\forall \mathbf{m} \in \mathbb{R}^m, \; \mathbf{m}^T \mathbf{A}\mathbf{m} \geq \mathbf{m}^T \mathbf{B}\mathbf{m}$. Furthermore, the largest and the smallest eigenvalues of $\mathbf{A}$ are no smaller than those of $\mathbf{B}$.*

*Proof.* Because $\forall \, i, a_i \geq b_i$, we have $\forall \mathbf{m} \sum_{i=1}^N (a_i - b_i)(\mathbf{x}_i^T \mathbf{m})^2 \geq 0$, which can be re-organized into $\mathbf{m}^T \mathbf{A}\mathbf{m} \geq \mathbf{m}^T \mathbf{B}\mathbf{m}$.

$\mathbf{A}$ is a symmetric matrix, so the largest eigenvalue $\lambda_1(\mathbf{A})$ is $\max_{\|\mathbf{m}\|_2=1} \mathbf{m}^T \mathbf{A}\mathbf{m} = \max_{\|\mathbf{m}\|_2=1} \sum_{i=1}^N a_i(\mathbf{x}_i^T \mathbf{m})^2$. Similarly, we have $\lambda_1(\mathbf{B}) = \max_{\|\mathbf{m}\|_2=1} \sum_{i=1}^N b_i(\mathbf{x}_i^T \mathbf{m})^2$. Let $\mathbf{m_B} \in \arg\max_{\|\mathbf{m}\|_2=1} \sum_{i=1}^N b_i(\mathbf{x}_i^T \mathbf{m})^2$. Then we have

$$\lambda_1(\mathbf{B}) = \sum_{i=1}^N b_i(\mathbf{x}_i^T \mathbf{m_B})^2 \leq \sum_{i=1}^N a_i(\mathbf{x}_i^T \mathbf{m_B})^2 \leq \max_{\|\mathbf{m}\|_2=1} \sum_{i=1}^N a_i(\mathbf{x}_i^T \mathbf{m})^2 = \lambda_1(\mathbf{A}) .$$
(26)

In the same way as for the largest eigenvalue, the smallest eigenvalue of $\mathbf{A}$ and $\mathbf{B}$ are $\lambda_m(\mathbf{A}) = \min_{\|\mathbf{m}\|_2=1} \sum_{i=1}^N a_i(\mathbf{x}_i^T \mathbf{m})^2$ and $\lambda_m(\mathbf{B}) = \min_{\|\mathbf{m}\|_2=1} \sum_{i=1}^N b_i(\mathbf{x}_i^T \mathbf{m})^2$, respectively. Let $\mathbf{m_A} \in \arg\min_{\|\mathbf{m}\|_2=1} \sum_{i=1}^N a_i(\mathbf{x}_i^T \mathbf{m})^2$. Then we have

$$\lambda_m(\mathbf{A}) = \sum_{i=1}^N a_i(\mathbf{x}_i^T \mathbf{m_A})^2 \geq \sum_{i=1}^N b_i(\mathbf{x}_i^T \mathbf{m_A})^2 \geq \min_{\|\mathbf{m}\|_2=1} \sum_{i=1}^N b_i(\mathbf{x}_i^T \mathbf{m})^2 = \lambda_m(\mathbf{B}) .$$
(27)

$\square$

Let us now go back to Theorem 3.

*Proof.* We first calculate the first and second derivatives of $\mathcal{L}_\epsilon(\mathbf{w})$ in Equation 6, as

$$\begin{aligned}
\nabla_\mathbf{w} \mathcal{L}_\epsilon(\mathbf{w}) &= \frac{1}{N}\sum_{i=1}^N -\frac{1}{1 + e^{y_i \mathbf{w}^T \mathbf{x}_i - \epsilon}} y_i \mathbf{x}_i , \\
\nabla_\mathbf{w}^2 \mathcal{L}_\epsilon(\mathbf{w}) &= \frac{1}{N}\sum_{i=1}^N \frac{e^{y_i \mathbf{w}^T \mathbf{x}_i - \epsilon}}{(1 + e^{y_i \mathbf{w}^T \mathbf{x}_i - \epsilon})^2} y_i^2 \mathbf{x}_i \mathbf{x}_i^T = \frac{1}{N}\sum_{i=1}^N \frac{e^{y_i \mathbf{w}^T \mathbf{x}_i - \epsilon}}{(1 + e^{y_i \mathbf{w}^T \mathbf{x}_i - \epsilon})^2} \mathbf{x}_i \mathbf{x}_i^T .
\end{aligned}$$
(28)

The second equality of $\nabla_\mathbf{w}^2 \mathcal{L}_\epsilon(\mathbf{w})$ is satisfied because $y_i$ is either $+1$ or $-1$. The dataset $\{(\mathbf{x}_i, y_i)\}_{i=1}^N$ is linearly separable under adversarial budget $\mathcal{S}_{\hat{\epsilon}}^{(p)}(\mathbf{x})$, so, $\forall i, \; y_i \mathbf{w}^T \mathbf{x}_i \geq \hat{\epsilon}$. When $\epsilon \leq \hat{\epsilon}$, $e^{y_i \mathbf{w}^T \mathbf{x}_i - \epsilon} > 1$ and monotonically decreases with $\epsilon$. As a result, $\frac{e^{y_i \mathbf{w}^T \mathbf{x}_i - \epsilon}}{(1 + e^{y_i \mathbf{w}^T \mathbf{x}_i - \epsilon})^2}$ monotonically increases with $\epsilon$ in the range $[0, \hat{\epsilon}]$.

Based on Lemma 1, $\forall \mathbf{m} \in \mathbb{R}^m, \; \mathbf{m}^T \nabla_\mathbf{w}^2 \mathcal{L}_\epsilon(\mathbf{w})\mathbf{m}$ increases with $\epsilon$, and so do the largest and the smallest eigenvalues of the Hessian matrix $\nabla_\mathbf{w}^2 \mathcal{L}_\epsilon(\mathbf{w})$.

$\square$

## C  Additional Experiments

### C.1  Experimental Details

For MNIST, we set the step size of PGD to $0.01$ and the number of iterations to $\epsilon/0.01 + 10$. For CIFAR10, we set the number of PGD iterations to 10 and the step size to $\epsilon/4$. The network architectures we use are the same as the ones in [52]. We provide the details in Table 2 and use a factor $w$ to control the width of the network. Unless specified, the LeNet models on MNIST have a width factor of 16, the VGG and ResNet18 models on CIFAR10 have a width factor of $8$.

| Name | Architecture |
|---|---|
| MNIST, LeNet | Conv($2w$), Conv($4w$), FC($196w$, $64w$), FC($64w$, 10) |
| CIFAR, VGG | Conv($4w$) $\times$ 2, M, Conv($8w$) $\times$ 2, M, Conv($16w$) $\times$ 3, M Conv($32w$) $\times$ 3, M, Conv($32w$) $\times$ 3, M, A, FC($32w$, 10) |
| CIFAR, ResNet18 | ResNet18 in [23], which uses a width $w = 16$ |

Table 2: Network architectures. Conv, FC, M and A represent convolutional layers, fully-connected layers, max-pooling layers and average pooling layers, respectively. The parameter of the convolutional layers indicates the number of output channels. The parameters of the fully-connected layers indicate the number of input and output neurons. The kernel sizes of the max-pooling layers and average pooling layers are always 2. $w$ corresponds to the width factor mentioned in Section 4 of the main paper.

We train the models for 100 epochs on MNIST and 200 epochs on CIFAR10. Unless explicitly mentioned, for LeNet models on MNIST, we use Adam [28] with a learning rate of $1 \times 10^{-4}$. For VGG models on CIFAR10, we also use Adam, with an initial learning rate of $1 \times 10^{-3}$, decreased exponentially to $1 \times 10^{-4}$ between the 100th epoch and the 150th epoch, and then fixed to $1 \times 10^{-4}$ after 150 epochs. For ResNet18 models on CIFAR10, we use accelerated SGD with a momentum factor of $0.9$. The initial learning rate is $0.1$ and is divided by 10 after 100 and 150 epochs.

(a) CIFAR10, VGG, Vanilla    (b) CIFAR10, ResNet18, Vanilla    (c) CIFAR10, VGG, Periodic    (d) CIFAR10, ResNet18, Periodic

Figure 6: Learning rate scheduling for VGG-8 and ResNet18-8 for CIFAR10 classification.

**Experiments in Section 5.**    The details of the adversarial attacks used in this section are demonstrated below:

- PGD: for MNIST models, PGD with 50 iterations, the step size is $0.1$; for CIFAR10 models, PGD with 10 iterations, the step size is $2/255$.
- PGD100: PGD with 100 iterations, the step size is $0.1$ for MNIST models and $1/255$ for CIFAR10 models.
- APGD100-CE: AutoPGD with 100 iterations and cross-entropy loss. We use the default settings in [10], i.e., $\rho = 0.75$, $\alpha = 0.75$.
- APGD100-DLR: AutoPGD with 100 iterations and difference-of-logit-ratio loss. We use the default settings in [10], i.e., $\rho = 0.75$, $\alpha = 0.75$.
- Square5K: SquareAttack with 5000 iterations. We use the default settings in [3], i.e., we use the *margin loss*.

For the results in Table 1, we fine-tune the weight-decay factor, choosing $1 \times 10^{-3}$ as the optimal value. In periodic settings, the learning rate and the adversarial budget are reset after 100 and 150 epochs. The scheduling in each period is scaled proportionally. We plot the learning rate scheduling

curves for VGG-8 and ResNet18-8 in Figure 6 for both the vanilla and periodic settings. Regarding the scheduling of $\epsilon$, we do not fully explore the value range of the hyper-parameters in the cosine and linear schedulers. We use $\epsilon_{min} = 0$ for all experiments. For the MNIST experiments, we set $\epsilon_{max} = 0.6$ for the cosine scheduler and $\epsilon_{max} = 0.8$ for the linear one. For the CIFAR10 experiments, we set $\epsilon_{max} = 16/255$ for both the cosine and linear schedulers. We plot the curves for $\epsilon_{cos}(d)$ and $\epsilon_{lin}(d)$ in Figure 7.

(a) MNIST, Cosine     (b) CIFAR10, Cosine     (c) MNIST, Linear     (d) CIFAR10, Linear

Figure 7: Adversarial budget scheduling for MNIST and CIFAR10 models.

## C.2 Additional Experimental Results

### C.2.1 Additional Results for Section 4.1

To complement the results on CIFAR10 models in Section 4.1, in Figure 8, we provide a numerical analysis on the first and last 500 training mini-batches of MNIST under different values of $\epsilon$. We observe the same phenomenon: in the early stages of training, a large adversarial budget leads to smaller gradient magnitudes and slows down the training; in the final stages of training, a large adversarial budget yields severe gradient scattering, indicated by larger gradient magnitudes.

(a) $\|\nabla_\theta \widehat{\mathcal{L}}_\epsilon(\theta)\|$, first 500.    (b) $\mathcal{E}_\epsilon(\theta)$, first 500.    (c) $\|\theta - \theta_0\|$, first 500.    (d) $\|\nabla_\theta \widehat{\mathcal{L}}_\epsilon(\theta)\|$, last 500.

Figure 8: Norm of the stochastic gradient $\|\nabla_\theta \widehat{\mathcal{L}}_\epsilon(\theta)\|$, robust training error $\mathcal{E}_\epsilon(\theta)$, distance from the initial point $\|\theta - \theta_0\|$ during the first or last 500 mini-batch updates for MNIST models.

### C.2.2 Additional Results for Section 4.2

Figure 9: Top 20 eigenvalues of the Hessian matrix for LeNet models. Both normalized (solid) and original (dashed) values are shown.

Figure 10: Cosine similarity between perturbations $\mathbf{x}'_{a\mathbf{v}} - \mathbf{x}$ and $\mathbf{x}'_{-a\mathbf{v}} - \mathbf{x}$. $\mathbf{v}$ can be either the top eigenvector (dashed) or randomly picked (solid).

Figure 11: The learning curves of training loss for CIFAR10 models (left) and MNIST models (right) under different values of $\epsilon$.

In Figure 9, we provide the Hessian spectrum analysis for LeNet models on MNIST under various adversarial budgets. As in Figure 3, the top eigenvalues, both the original and normalized values, of the Hessian matrix of our trained models are larger in the presence of larger adversarial budgets.

Note that the magnitudes of $\mathcal{L}_\epsilon(\theta)$ with different $\epsilon$ are similar. To show this, we randomly sample 10 $\theta$ and calculate the value of $\mathcal{L}_\epsilon(\theta)$ over the training set. For CIFAR10 models, the mean values are 2.3034, 2.3044, 2.3053, 2.3071 when $\epsilon$ is 0, 2/255, 4/255 and 8/255, respectively. For MNIST models, the mean values are 2.3029, 2.3414, 2.3424, 2.3429, 2.3432 when $\epsilon$ is 0, 0.1, 0.2, 0.3 and 0.4, respectively. In Figure 11, we plot the learning curves of the training loss; this clearly shows that the magnitudes of $\mathcal{L}_\epsilon(\theta)$ during training with different $\epsilon$ values are similar. Furthermore, the range of values of $\mathcal{L}_\epsilon(\theta)$ during training is smaller under large values of $\epsilon$. As a result, the increased curvature under large adversarial budgets is not caused by the magnitudes of the function $\mathcal{L}_\epsilon(\theta)$, and we empirically observed that optimization in adversarial training cannot be facilitated by tuning the learning rate.

Ideally, the sharpness of the minima is depicted by the condition number of its Hessian matrix. However, in the context of deep neural networks, the eigenvalue with the smallest absolute value of the Hessian matrix is almost zero, which renders the computation of the condition number both algorithmically and numerically unstable [16]. Instead, the spectral norm and the nuclear norm of the Hessian matrix are typically used as quantitative metrics for the sharpness of the minima [11]. Figure 3 and Figure 9 thus demonstrate that the obtained minima are shaper when $\epsilon$ is larger.

(a) MNIST, $\epsilon = 0.0$    (b) MNIST, $\epsilon = 0.1$    (c) MNIST, $\epsilon = 0.3$    (d) MNIST, $\epsilon = 0.4$

(e) CIFAR10, $\epsilon = 0$    (f) CIFAR10, $\epsilon = 2/255$    (g) CIFAR10, $\epsilon = 4/255$    (h) CIFAR10, $\epsilon = 8/255$

Figure 12: Loss landscape $\mathcal{L}_\epsilon(\theta + a_1\mathbf{v}_1 + a_2\mathbf{v}_2)$ under different adversarial budgets. $\theta$, $\mathbf{v}_1$, $\mathbf{v}_2$ are the parameter, and the first and second unit eigenvectors of the Hessian matrix. (Note that the z-scale for $\epsilon = 0.4$ in the MNIST case differs from the others.)

In Figure 10, we report the cosine similarity of input perturbations when we move the model parameter $\theta$ in two opposite directions. As in Figure 4, we see high similarity of the perturbations and high robust accuracy when $\mathbf{v}$ is a random direction. By contrast, when $\mathbf{v}$ is the first eigenvector of the Hessian matrix, we see a sharp decrease in both the perturbation similarity and robust accuracy as the

value of $a$ increases. In Figure 10, we only plot the perturbation similarity when the robust accuracy on the training set is higher than $70\%$; otherwise the model parameters can no longer be considered to be in a small neighborhood of the original ones.

Figure 12 shows 3D visualizations of $\mathcal{L}_\epsilon(\theta)$ under different values of $\epsilon$ in the parameter neighborhood of our obtained MNIST and CIFAR10 models on the training set. We study the curvature in the directions of the top 2 eigenvectors. The curvature clearly increases with $\epsilon$ and the corresponding minima become sharper.

### C.2.3 Additional Results for Section 5

Here, we compare the performance of warmup in the adversarial budget and warmup in the learning rate. As in Figure 5, we use the LeNet model on MNIST and set the target adversarial budget size to $0.4$. Our warmup period consists of the first 10 epochs: the learning rate starts at $0$ and linearly increases to the final value in the warmup period; the learning rate remains constant after the warmup period. In Figure 13, we show the robust accuracy on the test set when the final learning rate is set to $1 \times 10^{-4}, 3 \times 10^{-4}$ and $1 \times 10^{-3}$. For comparison, we show the best performance obtained when using warmup in the adversarial budget with a blue line. We run each experiment 5 times.

When the final learning rate is $1 \times 10^{-4}$, the learning rate wamup performance is not as good as warmup in the adversarial budget. When the final learning rate is $3 \times 10^{-4}$ or $1 \times 10^{-3}$, the variance of the performance becomes large. Learning rate warmup can sometimes yield good performance but sometimes fails to converge.

Figure 13: Mean and standard deviation of the test error on MNIST models when we use learning rate warmup but constant adversarial budget. The best performance by constant learning rate but adversarial budget warmup is depicted by a blue line.

Figure 14: Mean and standard deviation of the test error with LeNet-16 models of different sizes on MNIST, using different adversarial budget scheduling schemes.

### C.2.4 Robustness v.s. Model Capacity

In Figure 14, we report the performance of LeNet models of different width factors $w$ using different schedulers for $\epsilon$. The adversarial budget size $\epsilon$ at test time is $0.4$. We set the learning rate in Adam to be $10^{-4}$, because, for constant $\epsilon$ during training, it yields the best performance. Both Cosine and Linear schedulers outperform using a constant $\epsilon$ in all cases. When the model size is small, e.g., $w = 4$ and $w = 2$, using a constant $\epsilon$ during training fails to converge, but the Cosine and Linear schedulers still yield competitive results.

### C.2.5 Connectivity of Different Minima

The minima reached in the loss landscape of vanilla training have been found to be well connected [12, 14]. That is, if we train two neural networks under the same settings but different initializations, there exists a path connecting the resulting two models in the parameter space such that all points along this path have low loss. In this section, we study the connectivity of different trained models in adversarial training. Similarly to [14], we parameterize the path joining two minima using a *general Bezier curve*. Let $\theta_0$ and $\theta_n$ be the parameters of two separately-trained models, and $\{\widehat{\theta}_i\}_{i=1}^{n-1}$ the

(a) MNIST, training loss.

(b) MNIST, test error.

(c) CIFAR10, training loss.

(d) CIFAR10, test error.

Figure 15: Training loss and test error along the path connecting the minima of two independently-trained models.

parameters of $(n-1)$ trainable intermediate models. Then, an $n$-order Bezier curve is defined as a linear combination of these $(n+1)$ points in parameter space, i.e.,

$$\mathcal{B}(t) = (1-t)^n\theta_0 + t^n\theta_n + \sum_{i=1}^{n-1}\binom{n}{t}(1-t)^{n-i}t^i\widehat{\theta}_i .\tag{29}$$

$\mathcal{B}(t)$ is a smooth curve, and $\mathcal{B}(0) = \theta_0$ and $\mathcal{B}(1) = \theta_n$. We train $\{\widehat{\theta}_i\}_{i=1}^{n-1}$ by minimizing the average loss along the path: $\mathbb{E}_{t\sim U[0,1]}\mathcal{L}_\epsilon(\mathcal{B}(t))$, where $U[0,1]$ is the uniform distribution between 0 and 1. We use the Monte Carlo method to estimate the gradient of this expectation-based function and minimize it using gradient-based optimization. We use second-order Bezier curves to connect MNIST model pairs and fourth-order Bezier curves to connect CIFAR10 model pairs. When evaluating the models on the learned curves, we re-estimate the running mean and variance in the batch normalization layer based on the training set. The results are reported based on the evaluation mode of the models, and we turn off data augmentation to avoid stochasticity.

In Figure 15, following [14], we plot the training loss and test error along the learned curve, as a function of $t$ in Equation (29). For vanilla training or when the adversarial budget is small, we can easily find flat curves connecting different minima. However, the learned curves are not flat anymore when the adversarial budget increases. This indicates that the minima are less well-connected under adversarial training, and that it is more difficult for the optimizer to find a minimum.