[Reviews · NeurIPS 2020]

Review 1

Summary and Contributions: The authors conducted a novel analysis of adversarial training. In particular, they showed that adversarial training with a large perturbation budget results in gradient scattering and slow convergence. The results are intuitive yet, novel. To address this issue, the authors proposed using a periodic learning rate schedule, a known technique that helps optimization in vanilla settings. The experiments showed that the periodic schedule leads to non-trivial improvement in robustness over adversarial training.

Strengths: - A novel analysis of adversarial training technique; a theoretical explanation of why adversarial training with larger perturbation is unstable and converges slowly. - A non-trivial improvement of adversarial training using a periodic adversarial budget schedule.

Weaknesses: - As the authors have noted, the second-order gradient of the ReLU function is zero. The theory presented in this paper does not explain why gradient scattering happens in neural networks with ReLu activation functions. - Experimental evaluation should be improved. In the experiments, the authors used a PGD attack with 0.01 step size and eps/.01 + 10 iterations (MNIST) and 10 iterations and eps / 4 for CIFAR10. This is definitely non-standard parameters for the attack. The authors might consider fixing the number of steps and perform the grid search for the optimal step size to find the best attack for each epsilon. In addition, to ensure that gradient masking is not happening, the authors should add evaluation with 100 random restarts and evaluation using black-box attacks.

Correctness: The claims are correct. However, the empirical methodology can be improved. In particular: 1) The developed theory does not explain gradient scattering for ReLU networks. 2) The evaluation of the proposed defense might be flawed because: a) PGD attack parameters are non-standard; b) no extensive grid search was performed to find the optimal parameters of the attack; c) no random restarts was used; d) no evaluation on black-box attacks to confirm that gradient masking/shattering is not happening. 3) On MNIST / CIFAR10 the defense should be compared against standard adversarial training models from mnist_challenge / cifar10_challenge.

Clarity: Yes. The paper is well written. All theoretical and empirical results are clearly presented.

Relation to Prior Work: Yes. The paper discussed the related work in detail, but I might be unaware of some relevant works.

Reproducibility: Yes

Additional Feedback: - Could you include a detailed evaluation of the proposed attack using PGD attack after grid search for the optimal step size? The computation budget (number of steps) can be fixed to 100 or 1000 iterations. - Could you include results for PGD with random restarts (100 or 1000)? - Could you include results for black-box attacks? For example, using Simple Black-Box attack. - Could you add results for l2 robustness? - Could you compare the proposed defense against standard defense from mnist/cifar challenges. ----- The authors thoroughly improved the experimental evaluation (using stronger white-box and black-box attacks) and I am raising my score to accept.


Review 2

Summary and Contributions: This paper studies the loss landscape of PGD-based adversarial training under the cross-entropy loss and compares this landscape with that of vanilla training of neural networks. They find that the landscape of adversarial training may be less favorable to optimization as the size of the allowable perturbation increases; to do so, they investigate linear models and they also analyze properties of SGD, which is commonly used to train such networks in the deep, nonlinear setting. Further, they propose a learning rate schedule and a method of chunking the training process to predict using an ensemble of models to lower the error rate in adversarial training.

Strengths: + Figure 5 is quite illustrative following Prop 2. I think it should go in the main text. This would much better help to bring across this "scattering" phenomenon, which I didn't fully get until I saw that picture. + This study is well-motivated. Indeed many works do look at the landscape of vanilla training, but much less is known about the landscape of adversarial training. + The linear analysis provides some interesting insights. In particular, it seems intuitive that we should first be able to explain the behavior of linear classifiers before understanding the more complex case of deep classifiers. The claim that linear classifiers tend to become constant for large enough adversarial budgets is quite interesting and should be of interest to the growing community of folks who are trying to understand the properties of adversarial training by studying the linear case. + This "gradient scattering" phenomenon is quite interesting and not necessarily intuitive. It should be of independent interest to the community and after reading it seems to me that this is the foremost result in the paper. I would like to see more work done here to further explain this phenomenon. + The experiments are relatively thorough and consider several well-known datasets and architectures.

Weaknesses: - What is the definition of error and robust error? Is it just 1-accuracy or something else? - Proposition 1 and its proof are not clear to me. In particular, the paragraph after Prop. 1 is confusing. You haven't yet even defined g_\eps(x, W), and it's unclear to me why the version space V_\eps is defined in terms of g_\eps(x,W). Can't we simply define the version space as a function of x and W rather than including the nebulously defined function g_\eps? Further, the "proof" seems strange to me. Why not just say the following: let \eps_1\leq\eps_2. Now let \tilde{W} be any element of V_{\eps_2}. Then \tilde{W} is robustly optimal for all x' \in S_{\eps_2}(x). Since by assumption \eps_2\geq\eps_1, it follows that S_{\eps_1} \subset S_{\eps_2}. Therefore, \tilde{W} is robustly optimal for all x''\in S_{\eps_1} and thus \tilde{W}\in V_{\eps_1}. Therefore, V_{\eps_2} \subset V_{\eps_1}. QED. This seems much more straightfoward and even more general than the "easy" proof that was presented. For example, while it seems true that g_\eps(x,W) <= log K due to the choice of the cross-entropy loss, but with the above proof you don't even need - The authors claim that T_\eps is the complement of V_\eps in binary classification. No proof is given and no previous work is cited. It's not obvious upon reading this why this claim is immediately true and not worthy of further comment. - On page 4, I am confused by the assumption "we assume different gradients of the model parameters for different inputs." Is this something that is valid to assume? It seems to be assuming the conclusion that you want, which is that \lim_{\theta\rightarrow\theta_1} \nabla_\theta g_\eps(x,\theta) \neq \nabla_\theta g_\eps(x,\theta_1). - To get the gradient discontinuity property that you are looking for after Prop2, I think you should appeal to the fact that the second bound is _tight_. If it were not tight, then the conclusion of non-smoothness would not necessarily hold. (I see in the appendix that it is in fact tight, but the main text should appeal to this point.) - The idea of introducing a warm-up scheduling is interesting, but based on Table 1 it seems that it doesn't have a very significant effect. In particular, on CIFAR10 with VGG, it's improvement over a constant LR schedule is minimal. And in the other cases the improvements are on the order of 1-2%. Thus I think the authors overclaim slightly when they say that their strategy "can effectively overcome these challenges." - It would have been nice to have some experiments with linear classifiers on synthetic data just to verify empirically the results in Theorem 1. - The Hessian sampling is interesting, but it would nicer is there was something that could give guarantees for a local Lipschtiz constant. In particular, the works that consider bounding the output set of a neural network using e.g. interval bound propagation or convex relaxations may be more useful here.

Correctness: I looked through the proofs in the appendix and everything looked fine. I have more detailed comments about potential correctness issues in the "Weaknesses" section.

Clarity: The writing is relatively clear throughout. Aside from a few sections that I highlighted in the "weaknesses" section, the sections flow nicely from one page to the next. One place where the clarity could be improved is the following: The discussion after eq(3) is a bit confusing. In particular, the sentence that includes the lim_{\theta\rightarrow\theta_1} is hard to parse. It seems like the message here in plain words is that "gradients in small neighborhoods of theta-space can change discontinuously." It might make sense to try to use high-level ideas like this to make things a bit more clear.

Relation to Prior Work: In the related work, the authors claim that PGD is stronger than FGSM. This claim has been disputed recently in the work of Wong and Kolter. It is fair to say that while this hypothesis was initially accepted as true, it is no longer set in stone. The related work should be revised to reflect this. The authors also say that methods other than FGSM/PGD have been shown to over-regularize models and to significantly decrease clean accuracy. However, certainly the latter and probably also the former are true for PGD/FGSM, so this statement is slightly misleading. Otherwise, the related work seems appropriate.

Reproducibility: Yes

Additional Feedback: Overall, this was a solid paper. The results for linear and deep classifiers were quite interesting. There are a number of technical points that I hope the authors can address (see "Weaknesses" section). The adversarial scheduling portion of the paper seemed much weaker as did the corresponding experiments. I think the paper could be stronger if it focused more on these theoretical properties and the accompanying experiments. The related work should also be rewritten slightly to add clarity.


Review 3

Summary and Contributions: The authors provide a rigorous and formal mathematical analysis of the loss surfaces of adversarial training methods. Their analysis shows that adversarial training can fail to converge, and based on this finding they suggest a training schedule for adversarial training that can mitigate these failures.

Strengths: -The paper provides a rigorous mathematical analysis of the loss surfaces of adversarial training, which is a useful insight -The paper shows that there is mathematical proof that adversarial training can fail to converge -The paper suggests a good way of overcoming this using a periodic adversarial training schedule

Weaknesses: The only tests were performed on MNIST and CIFAR10 with two different networks (LeNet and VGG/Resnet18 respectively). It would have been more convincing to see how the method compares with different datasets using the same network to preclude the network structure from explaining any differences. Additionally, a larger and more realistic dataset like imagenet would have been more convincing. Post-rebuttal: A quick literature review gave several recent techniques performing adversarial training on imagenet, although they are designed specifically with efficiency in mind as an extension of standard adversarial training to make it feasible on this large dataset. Based on this and the general portion of the rebuttal, I am not sure my reviews are adequately addressed.

Correctness: The authors provide sufficient proof of their claims in the appendices and include source code which validate their experiments. The experiments empirically show what their theoretical analysis suggests.

Clarity: The paper is difficult to understand but it is a complex subject. This paper is written towards a technical audience and significant background knowledge of the subject is assumed.

Relation to Prior Work: This work extends a family of work that studies loss surfaces in deep learning by augmenting those theories with the constraint of adversarial training.

Reproducibility: Yes

Additional Feedback:


Review 4

Summary and Contributions: The paper investigates the loss landscape of adversarial training and shows theoretical results that the landscape is spiky and hard to train on. More specifically, the authors prove that, under Lipschitz continuity assumptions on loss and its gradients, the probability that the norm of gradients is large increases quadratically with the norm of adversarial perturbation. The empirical results show that even for activation functions like ReLU, the so-called "gradient scattering" phenomenon still exists. Additional analysis on Hessians show that it is indeed the gradients corresponding to the large eigenvalues in the Hessian that are responsible for the non-smoothness of adversarial training loss landscape. The paper further proposes to periodically reset adversarial budget so that the optimization process can be smoothed (in the amortized way). The empirical results show that this simple periodic budget schedule significantly improves the performance.

Strengths: 1. With simple assumptions, the authors prove that the optimization landscape is highly non-smooth and large gradients are likely to appear more frequently with larger norm of perturbation. And the authors show empirical evidence that the gradient scattering phenomenon happens. 2. The paper introduces a simple yet effective training schedule with almost no additional computational cost.

Weaknesses: 1. The authors only test the simplest attack method. It is not clear whether the method works against other attack methods. 2. A loss landscape visualization like [1] will better justify the theoretical claims on the landscape. [1] Hao Li, Zheng Xu, Gavin Taylor, and Tom Goldstein. Visualizing the loss landscape of neural nets. arXiv preprint arXiv:1712.09913, 2017.

Correctness: Yes.

Clarity: The paper is well written and easy to follow.

Relation to Prior Work: Yes.

Reproducibility: Yes

Additional Feedback: 1. What is the performance of your method without ensemble?

[Author Response · NeurIPS 2020]

| Task | $\epsilon$ Scheduler | Without Learning Rate Resets | | | | | With Periodic Learning Rate Resets | | | | |
|---|---|---|---|---|---|---|---|---|---|---|---|
| | | Clean Error (%) | Robust Error (%) | | | | Clean Error (%) | Robust Error (%) | | | |
| | | | PGD100 | APGD100 CE | APGD100 DLR | Square5K | | PGD100 | APGD100 CE | APGD100 DLR | Square5K |
| MNIST | Const | 1.56(17) | 10.86(143) | 15.18(155) | 14.70(136) | 19.58(45) | | | | | |
| LeNet | Cosine | 1.08(2) | 8.46(82) | 14.36(134) | 13.46(129) | 16.78(25) | | - | | | |
| $\epsilon = 0.4$ | Linear | 1.06(6) | 8.79(116) | 13.91(150) | 13.17(120) | 17.05(45) | | | | | |
| CIFAR10 | Const | 28.25(47) | 56.19(32) | 58.18(46) | 58.65(69) | 54.37(29) | 28.33(81) | 54.16(26) | 55.45(26) | 56.56(4) | 52.85(18) |
| VGG | Cosine | 25.06(19) | 56.00(42) | 57.83(45) | 58.88(16) | 53.95(15) | 23.91(21) | 53.10(18) | 54.44(16) | 55.80(24) | 51.41(37) |
| $\epsilon = 8/255$ | Linear | 23.56(95) | 55.88(5) | 57.74(16) | 58.39(18) | 53.66(24) | 21.88(33) | 52.97(17) | 54.32(17) | 55.63(17) | 51.28(4) |
| CIFAR10 | Const | 18.62(6) | 54.97(9) | 57.26(13) | 56.60(25) | 50.59(19) | 21.00(5) | 48.87(25) | 50.29(27) | 50.98(6) | 46.84(9) |
| ResNet18 | Cosine | 18.43(26) | 53.85(21) | 56.16(18) | 55.77(24) | 49.60(18) | 19.90(18) | 48.49(27) | 49.71(22) | 50.54(9) | 46.19(11) |
| $\epsilon = 8/255$ | Linear | 18.55(14) | 53.41(10) | 55.69(17) | 55.45(22) | 49.66(28) | 20.26(28) | 48.52(13) | 49.73(9) | 50.68(11) | 46.47(26) |

Table 1: Clean and robust error on the test set under various adversarial attacks. The numbers between the brackets indicate the standard deviation across different runs. Specifically, for example, 28.25(47) stands for $28.25 \pm 0.47$.

We thank the reviewers for their constructive comments. Below, we first address the concerns raised by several reviewers
regarding the experimental evaluation, and then provide point-to-point responses to each reviewer.

The table above shows that **our PAS strategy still yields better performance under stronger attacks**. As suggested
by reviewer 1, we first evaluate our models using 100-iteration PGD with 10 restarts (PGD100). To solve the issue
of suboptimal step size, we also evaluate our models using the state-of-the-art AutoPGD attack [Croce ICML20],
which searches for the optimal step size. We run AutoPGD for 100 iterations, based on either the cross-entropy loss
(APGD100-CE) or the difference of logit ratio loss (APGD100-DLR). For black-box attacks, we run the state-of-the-art
SquareAttack [Andriushchenko ECCV20] for 5000 iterations (Square5K).

Furthermore, we would like to emphasize that the main focus of our work is optimiza-
tion. In this regard, **one main advantage of PAS is to avoid convergence failure and**
**make the optimization less sensitive to the learning rate**, as shown in Figure 4. The
figure on the right shows the same observation with $l_2$ attacks. Specifically, under an
$l_2$ adversarial budget $\epsilon = 4$ on MNIST (as in [Madry ICLR18]), adversarial training with
a constant $\epsilon$ fails to converge when using a high learning rate in Adam, e.g., $10^{-3}$. By
contrast, a linear/cosine scheduler is more robust to the choice of learning rate and
yields consistently better performance.

**Reviewer #1:** The requested additional experiments are presented above. _ReLU networks:_ We discuss ReLU networks
in detail in Appendix A.3. Corollary 1 shows that gradient scattering is more severe under larger $\epsilon$ for ReLU networks.
Gradient scattering is measured as the first-order gradient difference, i.e., $\|\nabla_\theta \mathcal{L}_\epsilon(\theta_1) - \nabla_\theta \mathcal{L}_\epsilon(\theta_2)\|$, whose upper bound
increases with $\epsilon$ for any activation function. _Experimental settings:_ The step size of PGD follows [Ye ICCV19] and is
the same as in the seminal work [Madry ICLR18]. _Comparison with challenges:_ Both challenges are leaderboards for
attacks, not defenses. The architecture of our MNIST models are the same as the ones in the challenge. When $\epsilon = 0.3$,
our model has better robust accuracy than the standard defense model provided in this challenge: our model yields a
robust accuracy of 95.08% / 93.01% under PGD100 / PGD100 with 50 restarts, whereas the corresponding accuracy for
the provided model is 92.52% / 89.62%. We will release our trained model for public testing.

**Reviewer #2:** More experiments are provided in the general response. _Error definition:_ Yes, they are 1-accuracy.
_Proposition 1:_ Following the definition in Equation 1, $g_\epsilon(\mathbf{x}, \mathbf{W})$ is the adversarial loss of the point $\mathbf{x}$ under the
adversarial budget $\mathcal{S}_\epsilon(\mathbf{x})$. We will revise the proof to make it clearer. _$\mathcal{V}_\epsilon$ and $\mathcal{T}_\epsilon$ in binary cases:_ Thanks for pointing
this out. We realized that this claim is incorrect and will remove it. However, this does not affect our other claims.
_Assumption on page 4:_ Precisely, our assumption is $\nabla_\theta g(\mathbf{x}_1, \theta) \neq \nabla_\theta g(\mathbf{x}_2, \theta)$ when $\mathbf{x}_1 \neq \mathbf{x}_2$. This assumption is
based on the _clean_ loss function $g$ and is true in general for deep neural networks. By contrast, the conclusion
involves the adversarial loss, and refers to the discontinuity of the parameter gradients, not the fact of having different
parameter gradients for different inputs. _Tightness of Proposition 2's bound:_ We will move this claim to the main text.
_Linear model experiments:_ We will add this to validate the theorem. _IBP-based local Lipschitz constant:_ IBP or convex
relaxation calculate the upper bound $U_\epsilon$ and the lower bound $I_\epsilon$ of the adversarial loss: $I_\epsilon(\mathbf{x}, \theta) \leq g_\epsilon(\mathbf{x}, \theta) \leq U_\epsilon(\mathbf{x}, \theta)$.
These bounds are computed based on the adversarial budget defined in the input space, whereas the Lipschitz constant
is defined in the parameter space. Therefore, the curvature of these bounds, i.e., $\nabla_\theta^2 I_\epsilon$ and $\nabla_\theta^2 U_\epsilon$, does not provide
a bound for $\nabla_\theta^2 g_\epsilon$. We believe that calculating a tight guaranteed bound of the Hessian eigenvalues is non-trivial.
_Clarity:_ Thanks, we will revise as suggested. _Related work:_ Note that we cite [Wong ICLR20], but their method differs
from vanilla FGSM [Goodfellow ICLR14] by using a random starting point, while FGSM uses a fixed one. Regarding
over-regularization, we agree that adversarially-trained models have considerably lower clean accuracy than the ones
trained using the clean data. However, as observed in [Zhang ICLR20], the models trained by some provably-robust
methods, such as convex relaxations, have even lower clean accuracy than those trained by PGD. We will clarify this.

**Reviewer #3:** We have included more results in the general response. We choose datasets commonly used in the
literature to facilitate comparisons. We are not aware of any work reporting adversarial training results on ImageNet.

**Reviewer #4:** More experimental results are provided in the general response. We cite [Li NIPS17] and visualize the
loss landscape in Figure 12 of Appendix C.2.2. In contrast to [Li NIPS17], however, we explore the neighborhood in the
directions of the top two Hessian eigenvalues, which more clearly shows the sharpness of the loss landscape.

[Meta-Review · NeurIPS 2020]

Paper shows the loss landscape of adversarial training is spiky and hard to train on, with interesting theoretical and experimental contributions. The reviewers had many questions, some answered in the author response, and we encourage the authors to address these best they can when they revise the paper.